# The structural basis for deubiquitination by the fingerless USP-type effector TssM

Thomas Hermanns[1], Matthias Uthoff[2], Ulrich Baumann[2], Kay Hofmann[1]

**Intracellular bacteria are threatened by ubiquitin-mediated autophagy, whenever the bacterial surface or enclosing membrane structures become targets of host ubiquitin ligases. As a countermeasure, many intracellular pathogens encode deubiquitinase (DUB) effectors to keep their surfaces free of ubiquitin. Most bacterial DUBs belong to the OTU or CE-clan families. The betaproteobacteria *Burkholderia pseudomallei* and *Burkholderia mallei*, causative agents of melioidosis and glanders, respectively, encode the TssM effector, the only known bacterial DUB belonging to the USP class. TssM is much shorter than typical eukaryotic USP enzymes and lacks the canonical ubiquitin-recognition region. By solving the crystal structures of isolated TssM and its complex with ubiquitin, we found that TssM lacks the entire "Fingers" subdomain of the USP fold. Instead, the TssM family has evolved the functionally analog "Littlefinger" loop, which is located towards the end of the USP domain and recognizes different ubiquitin interfaces than those used by USPs. The structures revealed the presence of an N-terminal immunoglubulin-fold domain, which is able to form a strand-exchange dimer and might mediate TssM localization to the bacterial surface.**

## Introduction

In eukaryotic cells, posttranslational protein modification by ubiquitin regulates nearly every cellular pathway, including the defense against bacteria and other intracellular pathogens. Several Gram-negative bacterial species are taken up by host cells and are able—or even required—to proliferate within the cytosol of eukaryotic cells. Some bacteria proliferate within vesicular structures called "bacteria-containing vacuoles," which develop from the initial phagosome under the influence of bacterial effectors (Santos & Enninga, 2016). *Legionella, Coxiella,* and *Chlamydia* are among the bacterial genera that use this lifestyle. Other bacteria, such as *Shigella, Francisella,* and *Burkholderia,* usually escape the phagosomes and proliferate directly within the cytosol (Ray et al, 2009). *Salmonella* can proliferate both within vacuoles and exposed to the cytosol (Stevenin et al, 2019). Irrespective of their intracellular lifestyle, these bacteria are threatened by ubiquitin-induced autophagy pathways, which can attack the cytosolic bacteria directly or the membranous structure they are contained in (Jo et al, 2013; Gomes & Dikic, 2014; Kimmey & Stallings, 2016; Tripathi-Giesgen et al, 2021). As a countermeasure, many intracellular bacteria secrete deubiquitinase (DUB) effectors into the host cell, which help them evade ubiquitination and thus ubiquitin-based lysosomal targeting (Hermanns & Hofmann, 2019; Franklin & Pruneda, 2021). Typical intracellular bacteria encode one or two DUBs, often without much linkage specificity or a moderate preference for K63-linked chains (Hermanns & Hofmann, 2019). *Legionella pneumophila* is the only known bacterium with a massively expanded and diversified ubiquitin-effector repertoire (Wan et al, 2019; Hermanns et al, 2020; Shin et al, 2020; Warren et al, 2023). Eukaryotic deubiquitinases are mostly papain-fold cysteine proteases, which are usually subdivided into the six structural classes USP, UCH, OTU, Josephin, MINDY, and ZUFSP (Clague et al, 2019), recently joined by VTD as the seventh class (Erven et al, 2022). Most bacterial DUBs are either OTUs or belong to a protease class called "CE clan," which in eukaryotes is used for cleaving ubiquitin-like modifiers rather than ubiquitin itself (Pruneda et al, 2016; Hermanns & Hofmann, 2019; Schubert et al, 2020).

One of the few exceptions is the USP-type deubiquitinase TssM, which has been characterized in *Burkholderia pseudomallei* (Tan et al, 2010) and *Burkholderia mallei* (Shanks et al, 2009), the causative agents of melioidosis and glanders, respectively. USP-type enzymes are the most widespread DUB type in eukaryotes (Clague et al, 2019), but TssM is the only described bacterial effector related to this family. *Burkholderia* TssM is expressed inside the host cell and suppresses the innate immune system by preventing the activation of the NF-κB and type I interferon pathways (Shanks et al, 2009; Tan et al, 2010). The Name TssM refers to a type-VI secretion system, because the gene is physically linked to and transcriptionally co-regulated with the T6SS-1 and T3SS-3 gene clusters of *B. mallei* (Shanks et al, 2009; Burtnick et al, 2011). However, *Burkholderia* TssM is unrelated to TssM proteins from other bacteria, which are components of the type-VI secretion

---

[1]Institute for Genetics, University of Cologne, Cologne, Germany   [2]Institute of Biochemistry, University of Cologne, Cologne, Germany

Correspondence: t.hermanns@uni-koeln.de
Matthias Uthoff's present address is Bayer AG, Research & Development, Pharmaceuticals, Biologics Research, Wuppertal, Germany

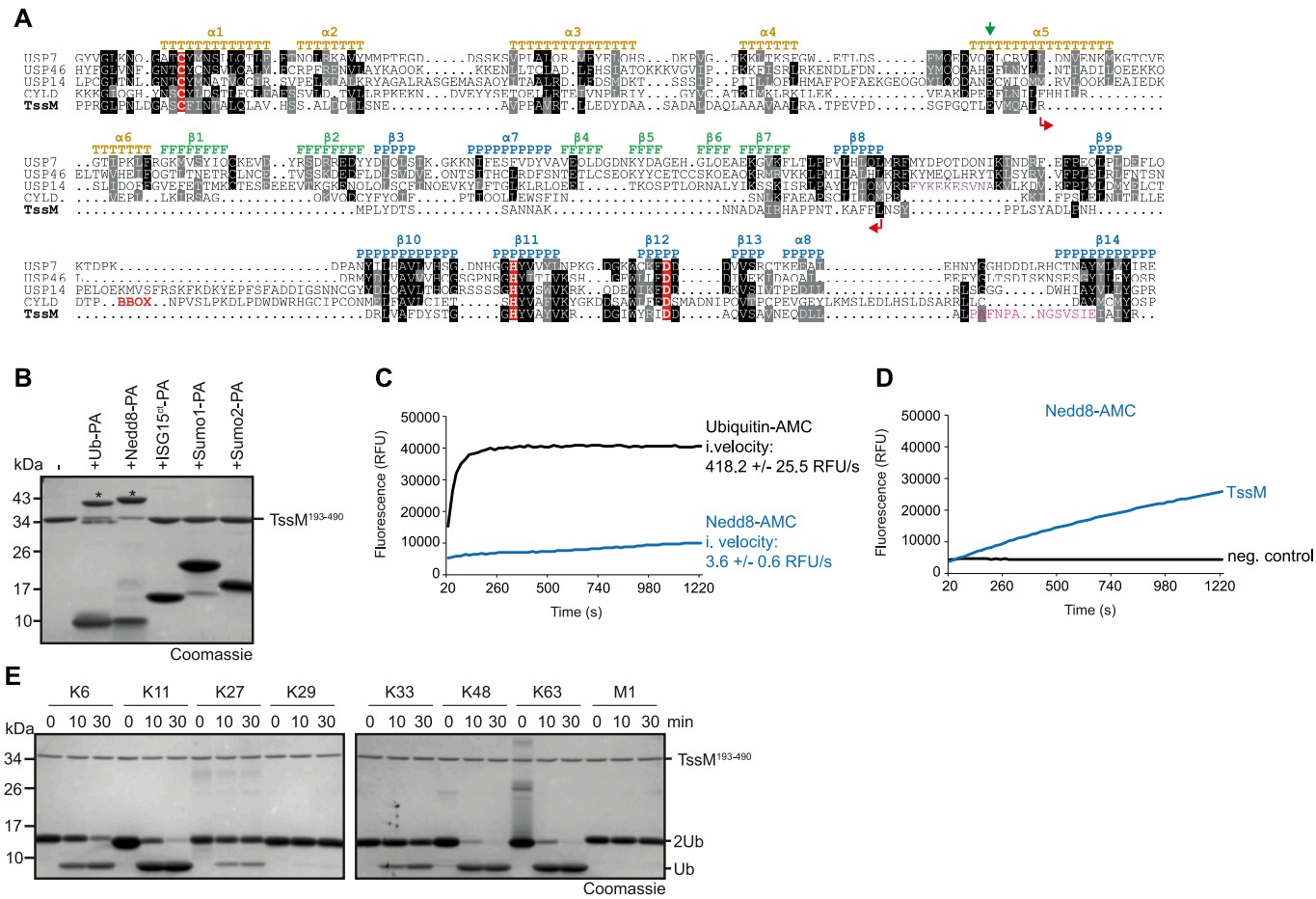

**Figure 1. TssM is a ubiquitin specific DUB.**
**(A)** Structure-guided alignment of TssM and representative members of the USP-family. Residues printed on black or grey background are invariant or conservatively replaced in at least 50% of the sequences. The active site residues are highlighted in red. Colored letters indicate the localization of the three USP subdomains Thumb (T), Fingers (F), and Palm (P) and the respective secondary structure elements of USP7. The internal TssM deletion is marked with red arrows. The sequence of the CYLD BBOX was removed from the alignment to get a better overview and the site is marked with a red BBOX. USP family members exhibit a highly conserved glutamic residue which is important for binding of ubiquitin's Arg72. This residue is marked by a green arrow. Blocking loop 1 of USP14 and Littlefinger loop of TssM are highlighted by purple and pink letters respectively. **(B)** Reaction of TssM with Ub and UbL activity-based probes. Asterisks mark the shifted bands after reaction. **(C, D)** Activity of 5 (C) or 50 nM (D) TssM against ubiquitin– and NEDD8–AMC (C) or against NEDD8–AMC (D). The RFU values are the means of triplicates. **(C)** The initial velocities (i. velocity) are the mean velocities derived from measurements shown in (C). **(E)** Linkage specificity analysis of TssM. A panel of homotypic di-ubiquitin chains was incubated with TssM for the indicated time points.
Source data are available for this figure.

machinery (Durand et al, 2015). Neither type-VI nor type-III secretion systems are required for the secretion of *Burkholderia* TssM, which rather appears to be a target of a type-II secretion system (Burtnick et al, 2014). Because intracellular *Burkholderia* bacteria escape the phagocytic membrane system (Allwood et al, 2011), type-II secretion should be sufficient to deliver effectors into the host cytosol (Korotkov et al, 2012). Whereas TssM is the only deubiquitinase described for *B. mallei* and *B. pseudomallei*, these pathogens encode a second ubiquitin-directed effector. ChbP acts as a ubiquitin deamidase that interferes with ubiquitin conjugation and thus impedes ubiquitin-based host defenses (Cui et al, 2010).

Although TssM is clearly related to eukaryotic USP-type deubi-quitinases, it is a very divergent member of this family and appears to lack sequence regions that are crucial for ubiquitin recognition in other USP enzymes (Fig 1A). Moreover, our recent investigation of the evolutionary relationship between cysteine protease families

(Hermanns et al, 2020) indicated distant similarities between the USP- and Josephin-type DUB families, which were most conspicuous for the USP-like subfamily containing TssM. In the present study, we analyzed the biochemical and structural bases of TssM deubiquitination activity, with the main aims to (i) determine whether TssM might be a missing link between USP and Josephin-type DUBs and (ii) to understand how TssM can recognize ubiquitinated substrates in the absence of the USP-typical ubiquitin-binding elements.

## Results

### TssM is a ubiquitin-specific, linkage promiscuous DUB

As a representative member of the TssM family, we decided to thoroughly characterize the TssM protein from *B. pseudomallei* with

respect to its ubiquitin recognition surface, catalytic properties, and structural relationship to different DUB families. When using sequence alignments of TssM and representative members of the USP family, clear similarities were observed, particularly around the conserved active site residues (Fig 1A). However, the alignment also shows a large internal deletion within the catalytic domain, which encompasses the entire "Fingers"-subdomain and some flanking regions. In eukaryotic USPs, this subdomain is responsible for recognizing the distal S1-ubiquitin (Hu et al, 2002). The TssM homologue from *B. mallei* was previously reported to be ubiquitin-specific, with some cross-reactivity towards NEDD8 (Shanks et al, 2009). Given that TssM from *B. mallei* and *B. pseudomallei* are >96% identical, this observation raises the question of how TssM can recognize ubiquitin in the absence of the interface usually provided by the Fingers subdomain.

Therefore, we tested the entire folded part of TssM from *B. pseudomallei* (TssM[193–490]) for UbL specificity, using a panel of activity-based probes that serve as substrate models for different ubiquitin-like modifiers (Fig S1A). As expected, TssM reacted with the ubiquitin and NEDD8 probes (Fig 1B). The more divergent UbLs ISG15, SUMO1, and SUMO2 did not react, indicating specificity for ubiquitin and Nedd8. A comparison of ubiquitin and Nedd8 reactivity using shorter time points revealed a faster reaction of the ubiquitin probe (Fig S1B). For a more quantitative comparison of TssM activity against ubiquitin and NEDD8, the liberation of an AMC fluorophore from C-terminally fused Ub-AMC and NEDD8-AMC model substrates was monitored. At an enzyme concentration of 5 nM, TssM fully cleaved ubiquitin-AMC within 3 min, whereas cleavage of NEDD8-AMC was barely detectable (Fig 1C). Upon a 10-fold increase in TssM concentration, cleavage of NEDD8-AMC became visible, although the reaction was not completed during the 20-min incubation period (Fig 1D). The comparison of the initial velocities of TssM cleaving ubiquitin-AMC (418.2 ± 25.5 RFU/s) and NEDD8-AMC (3.6 ± 0.6 RFU/s) indicates an ~116x faster cleavage of ubiquitin. Next, we tested TssM for linkage specificity using a panel of differently linked di-ubiquitin species. TssM mainly cleaved the K11, K48, and K63-linked chains. K6- K27- and K33-linked chains were poorly cleaved, whereas K29 and linear chains were not cleaved at all (Fig 1E). Taken together, TssM is—despite the apparent absence of the Fingers domain—a potent ubiquitin-specific isopeptidase, showing promiscuous cleavage of most ubiquitin chains.

## TssM assumes a divergent USP fold lacking the "Fingers" subdomain

To better understand the evolutionary position of TssM within the USP family and its possible relationship with the Josephin family, we solved the crystal structure of TssM[193–490] to a resolution of 3.15 Å (Table 1). This construct comprises the portion of TssM predicted to be structured, and upon crystallization revealed the presence of two globular folds: an N-terminal immunoglobulin-like domain, followed by the catalytic DUB domain (Fig 2A). The asymmetric unit contained two TssM molecules forming a possible dimer (Fig S2A) with a contact surface formed by the Ig-like domains, with some contributions by the catalytic domain. A single β-strand formed by the first eight residues of the Ig-like domain is swapped between the two adjacent dimers. The two chains are almost completely resolved, with the exception of residues 193 (chain B), 394, and 395 missing from both monomers, and residues 476–480 being disordered in chain B. Both chains have identical conformations and can be superimposed with an RMSD of 0.63 Å for 3,221 atoms (Fig S2B).

The DUB domain assumed a papain-like fold with an active site triad formed by Cys-308, His-442, and Asp-457 (Fig 2B and C). The role of these residues was confirmed by mutational analysis. As expected, C308A and H442A completely abrogated the cleavage of ubiquitin chains, whereas D457A showed strongly reduced cleavage (Fig 2D). In line with the sequence conservation (Fig 1A), the catalytic domain of TssM resembles USP-type deubiquitinases and can be successfully superimposed with the paradigmatic hand-shaped USP domain of USP7, which consists of the so-called Palm, Thumb, and Fingers subdomains (Hu et al, 2002). When matching the entire USP folds of TssM and USP7 (Fig S2C and D), the resulting RMSD of 5.4 Å is unusually high. This discrepancy is owed to the fact that the TssM structure lacks the entire Fingers subdomain, which is in accordance with the major deletion observed in the multiple sequence alignment (Fig 1A) within the respective region. The Palm and Thumb subdomains of the USP fold are conserved in TssM, but their relative orientation differs from canonical USP domains because of the absence of the Fingers region. This orientational difference explains the large overall RMSD, which considerably improves to 0.64 (167 atoms) or 2.15 Å (213 atoms), when superimposing only the Palm or Thumb regions, respectively (Fig S2E and F). The internal deletion within the TssM catalytic domain comprises not only the proper Fingers subdomain, but also two adjacent helices (α-6 and half of α-5 in USP7), which are part of the thumb region (Hu et al, 2002). In TssM, the entire deleted region is replaced by the short helix α6 flanked by flexible loops, which appear neither related to other USP domains, nor do they structurally correspond to canonical USP elements. Therefore, this region is not assumed to be a shortened Fingers remnant but rather a replacement to fill the gap resulting from the Fingers deletion. Structurally, this region lies within the (extended) palm subdomain. A further difference between TssM and USP7 is the shortened C-terminus. TssM ends on a β-strand corresponding to β14 of USP7 (Hu et al, 2002), whereas USP7 helices α9 and α10 do not have equivalents in TssM. However, these helices are a USP7-specific extension of the catalytic domain; the TssM c-terminus therefore corresponds to the canonical ending of the USP fold. In structural comparisons by Dali, the most similar structure was found to be the catalytic domain of USP46, followed by its close paralog USP12 (Holm, 2020). These two structures could be superimposed with TssM at RMSDs of 4.44 (USP46, 414 atoms) and 4.49 Å (USP12, 354 atoms), overall better than USP7 but still subject to improvement when aligning individual subdomains of the USP fold (Fig S2G).

The Ig-like domain is attached to the N-terminus of the catalytic domain via a flexible linker and is located on the opposite face of the active site. Therefore, we hypothesized that it does not influence the catalytic activity and generated a truncation lacking the Ig-like domain (TssM[292–490]). As expected, removal of the Ig-like domain did not affect the cleavage of ubiquitin–AMC or K63-linked di-ubiquitin (Fig 2E and F). This shows that the DUB domain alone is sufficient for cleavage, and suggests that the Ig-like domain fulfills another non-catalytic function.

**Table 1. Data collection and refinement statistics.**

| Structure | Apo | Ubiquitin complex |
|---|---|---|
| Light source | ESRF | SLS |
| Beam line | ID30B | X06SA |
| Detector | Pilatus6M | Eiger 16M |
| Number images | 1,700 | 1,800 |
| Wavelength | 0.91840 | 1.00000 |
| Cell dimensions ($\mathring{A}$) | 116.85, 116.85, 111.32 | 103.75, 103.75, 191.52 |
| Cell angles (deg.) | 90.00, 90.00, 120.00 | 90.00, 90.00, 90.00 |
| Space group | P 32 2 1 | P 41 21 2 |
| Resolution | 74.88 - 3.15 (3.23 - 3.15) | 103.74 - 1.62 (1.66 - 1.62) |
| Number unique reflections | 15,249 (1,109) | 132,701 (9,702) |
| Completeness (%) | 97.6 (96.9) | 99.9 (100.0) |
| Redundancy | 4.69 (4.76) | 13.42 (13.72) |
| CC1/2 | 98.5 (52.3) | 99.9 (51.6) |
| I/σ(I) | 5.75 (1.53) | 18.54 (1.19) |
| Rmeas (%) | 25.0 (107.9) | 8.0 (242.5) |
| Wilson B ($\mathring{A}^2$) | 58.24 | 28.87 |
| Reflections used in refinement | 15,239 | 130,701 |
| Resolution ($\mathring{A}$) | 50.60 - 3.15 | 91.22 - 1.62 |
| Rwork/Rfree (%) | 22.91/27.53 | 17.40/19.80 |
| Free R value test set size | 1,526 (10%) | 2,000 (1.5%) |
| Contents of asymmetric unit (proteins) | 2 * TssM[193–490] | 2 * TssM[193–490] + 2 * Ub-PA |
| Contents of asymmetric unit (ligands) | — | 2 * EDO, 1 * FLC, 2 * NA |
| No. of water molecules | 0 | 724 |
| No of atoms (without H) | 4,451 | 5,827 |
| Mean B ($\mathring{A}^2$) | 58.53 | 31.65 |
| Ramachandran outlier/favoured (%) | 0.17/96.53 | 0.0/97.96 |
| Rotamer outlier (%) | 0.21 | 0.31 |
| C-beta outlier (%) | 0.00 | 3 |
| Clashscore | 4.07 | 2.3 |
| RMSD angle/bond (deg./$\mathring{A}$) | 0.946/0.008 | 1.80/0.012 |

### TssM has a novel, USP-atypical, ubiquitin-binding interface

The specificity for ubiquitin implies the selective binding of ubiquitin to an S1 site on the surface of TssM. In eukaryotic USPs, the S1 site is typically formed by the Thumb and Fingers subdomains (Hu et al, 2002; Ye et al, 2009). Considering the fingerless structure of TssM, this raises the question of how the S1 site is formed. To answer this question, we reacted TssM[193–490] with ubiquitin–PA and solved the crystal structure of the covalent complex to a resolution

of 1.62 $\mathring{A}$ (Table 1) (Fig 3A). The asymmetric unit contained two monomeric copies of TssM ~ Ub-PA complexes, which could be superimposed with an RMSD of 0.488 $\mathring{A}$ (2,026 atoms) (Fig S3A). The catalytic domains with their covalently bound ubiquitin units are almost identical (RMSD 0.24 $\mathring{A}$), some variability is seen in the relative orientation of the Ig-like and catalytic domains (Fig S3B). TssM[193–490] was fully resolved, with the exception of a short linker region near the N-terminus (Pro-202 to Leu-207), which was not resolved in chain A. The comparison of the apo and the ubiquitin-bound form revealed no major conformational changes upon ubiquitin binding (Fig 3B); the DUB domains of the apo- and Ub-bound structures align with an RMSD of 0.35 $\mathring{A}$ (1,115 atoms). The only major difference between the two structures is the dimerization interface, which influences the positioning of the Ig-like domain and will be discussed below.

Owing to the missing Fingers region, the interaction interface between TssM and ubiquitin is relatively small. The contact surface (Vangone et al, 2011), has a size of 977 $\mathring{A}^2$, whereas USP7 and USP12 use ubiquitin recognition surfaces that are almost double in size (1,729 and 1,816 $\mathring{A}^2$ respectively). Nevertheless, ubiquitin has multiple contacts to the USP fold of TssM, mainly through the Palm subdomain. Most prominently, the linker region between α-7 and β-6 (residues 473–486) is in an optimal position to coordinate the S1-ubiquitin. Because of its analogous role to the missing Fingers region, we named this region Littlefinger (Fig 3A). Interestingly, the Littlefinger loop takes a similar position in the structure as the canonical blocking loop 1 of USP DUBs, although it is derived from a different part of the sequence (Figs 1A and S3C). The blocking loop was reported to contact ubiquitins Ile-36 patch (Hu et al, 2005), whereas the Littlefinger loop contacts hydrophobic residues around Ile-44 of ubiquitin. The Littlefinger residues Gly-480, Val-482, and Phe-475, together with Tyr418 (Thumb subdomain) form extensive hydrophobic interactions with the Ile-44 patch, but not Ile-44 itself (Fig 3C). Mutation of these residues to alanine led to a drastic reduction in ubiquitin chain and ubiquitin-AMC cleavage, highlighting the importance of this patch (Figs 3G and S3D). Members of the Josephin family do mainly contact ubiquitin's Ile44-patch as well. Therefore, we examined whether the contacting residues are conserved between TssM and ATX3L. However, superimposition of TssM and ATX3L showed that the bound ubiquitin molecules are rotated by 180°, indicating that although the same patch is used, binding of ubiquitin is achieved by completely different regions of the DUB (Fig S3E).

In addition to recognizing the Ile44-Patch, the Littlefinger loop stabilizes the ubiquitin C-terminus via hydrogen bonding to the main chain of Leu71 and Leu73 (Fig 3D). Eukaryotic USPs also stabilize these residues; however, they use a hydrophobic pocket to coordinate the leucine side chains. TssM further stabilizes the ubiquitin C-terminus through a hydrophobic interaction between Tyr-443 and the penultimate residue Gly-75 (Fig 3E). Tyr-443 of TssM represents the "aromatic motif," which is found in all eukaryotic DUB families (except ZUFSP) and is formed by the aromatic residue directly after the catalytic histidine; its importance for USP catalysis has been shown for the example USP21 (Hermanns et al, 2020). The alanine substitution of Tyr-443 leads to strongly reduced cleavage of di-ubiquitin and ubiquitin–AMC

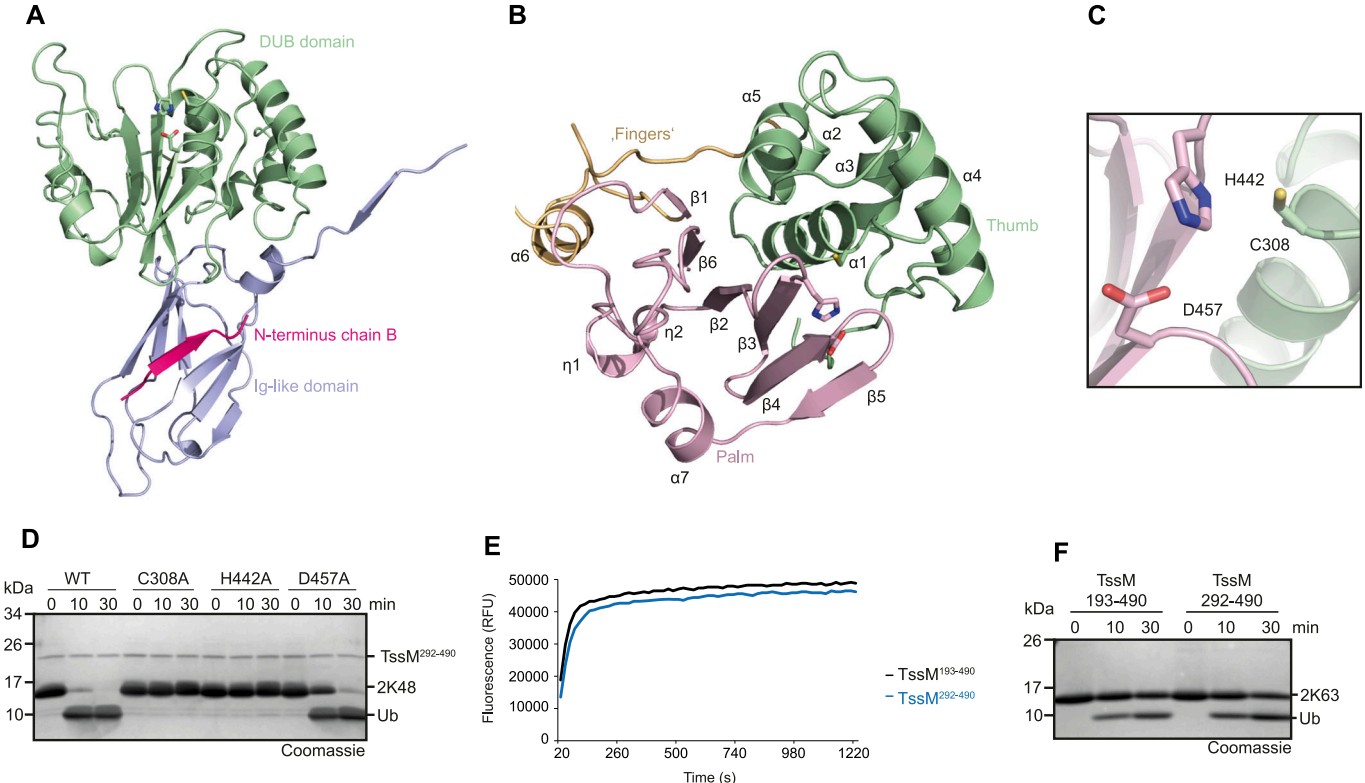

**Figure 2. TssM is a fingerless USP.**
**(A)** Crystal structure of TssM$^{193-490}$ shown in cartoon representation. The structure is divided in an Ig-like domain (light blue) and a DUB domain (green). The first $\beta$-strand undergoes a strand swap within the dimer. The respective $\beta$-strand of the other dimer is shown in cartoon representation and colored pink. The active site residues are shown as sticks. **(B)** The DUB domain of TssM adopts a USP-like fold. The USP-typical Thumb and Palm subdomains are conserved and colored green and light pink, respectively. The Fingers subdomain (yellow) is missing and replaced by a short $\alpha$-helix ($\alpha$-6). The active site residues are shown as sticks. **(C)** Magnification of the active site formed by Cys-308, His-442, and Asp-457. Cys-308 belongs to the Thumb subdomain and is colored green; His-442 and Asp-457 are located within the Palm Subdomain and colored light pink. **(D)** Activity of WT TssM compared with the active site mutants. 1.5 $\mu$M TssM were incubated with K48-linked di-ubiquitin and the reaction was stopped by the addition of Laemmli buffer after the indicated timepoints. **(E)** Activity of 5 nM TssM$^{193-490}$ and the Ig-lacking truncation TssM$^{292-490}$ against ubiquitin–AMC. The RFU values are the means of triplicates. **(F)** Activity of 0.5 $\mu$M TssM$^{193-490}$ and TssM$^{292-490}$ against K63-linked di-ubiquitin. The reaction was stopped by the addition of Laemmli buffer after the indicated time points, resolved by SDS–PAGE and Coomassie-stained.
Source data are available for this figure.

(Figs 3H and S3F), indicating that the aromatic motif also plays a crucial role in TssM. Recognition of the basic Arg-72/Arg-74 residues of ubiquitin by acidic DUB residues is highly conserved in USPs. Usually, Arg-72 forms a salt bridge with an invariant glutamate such as Glu-299 in USP7. Based on the sequence alignment, this residue is conserved as Glu-378 in TssM (Fig 1A). However, the Glu-378 side chain is oriented somewhat differently and forms a salt bridge with Arg-74 instead of Arg-72 (Fig 3F). Arg-72 is also stabilized by TssM but uses a non-conventional contact with Glu-485 in the Littlefinger region, supported by Gln-375 (Fig 3F). Individual alanine mutation of the Arg-72 contacting residues strongly reduced the cleavage of ubiquitin chains, with E485A being nearly inactive, highlighting the important role of this Littlefinger residue (Figs 3I and S3G). By contrast, the mutation of Glu-378 to alanine did not change ubiquitin–AMC cleavage and only mildly affected chain cleavage by TssM. (Figs 3I and S3G), indicating that the recognition of Arg-74 is not crucial for TssM. Additional hydrogen bonds were observed between Ser-391 and Asn-416 of the catalytic domain and Gln-49 of ubiquitin, which were shown to be mostly dispensable for catalysis (Fig S3H and I).

## Implications of the immunoglobulin-like domain

A major difference between the apo and ubiquitin-bound structures is the apparent dimerization state. Whereas the Ub-PA–bound structure contains two individual barely interacting TssM ~ Ub-PA complexes, the apo structure shows the Ig-like domains in a strand-swapped dimer, in which the first $\beta$-strand of each Ig-like domain folds with the remainder of the adjacent molecule (Fig 4A). Because the $\beta$-strand swap might stabilize a possible dimer in solution, we determined the approximate size of TssM$^{193-490}$ in solution by comparing it with calibration proteins during size-exclusion chromatography. In this experiment, TssM ran as a single peak, in between myoglobin (17.8 kD) and ovalbumin (44 kD), which is in line with the calculated molecular mass and suggests a monomeric state in solution (Fig 4B).

Apart from the strand swap, the Ig-like domains of both structures (and both chains) are structurally very similar; they can be superimposed with RMSDs between 1.1 and 1.4 Å. Beyond $\beta$1, the only additional difference is a short helix ($\alpha$1, residues 202–208) between the first two $\beta$-strands in the apo structure. This segment

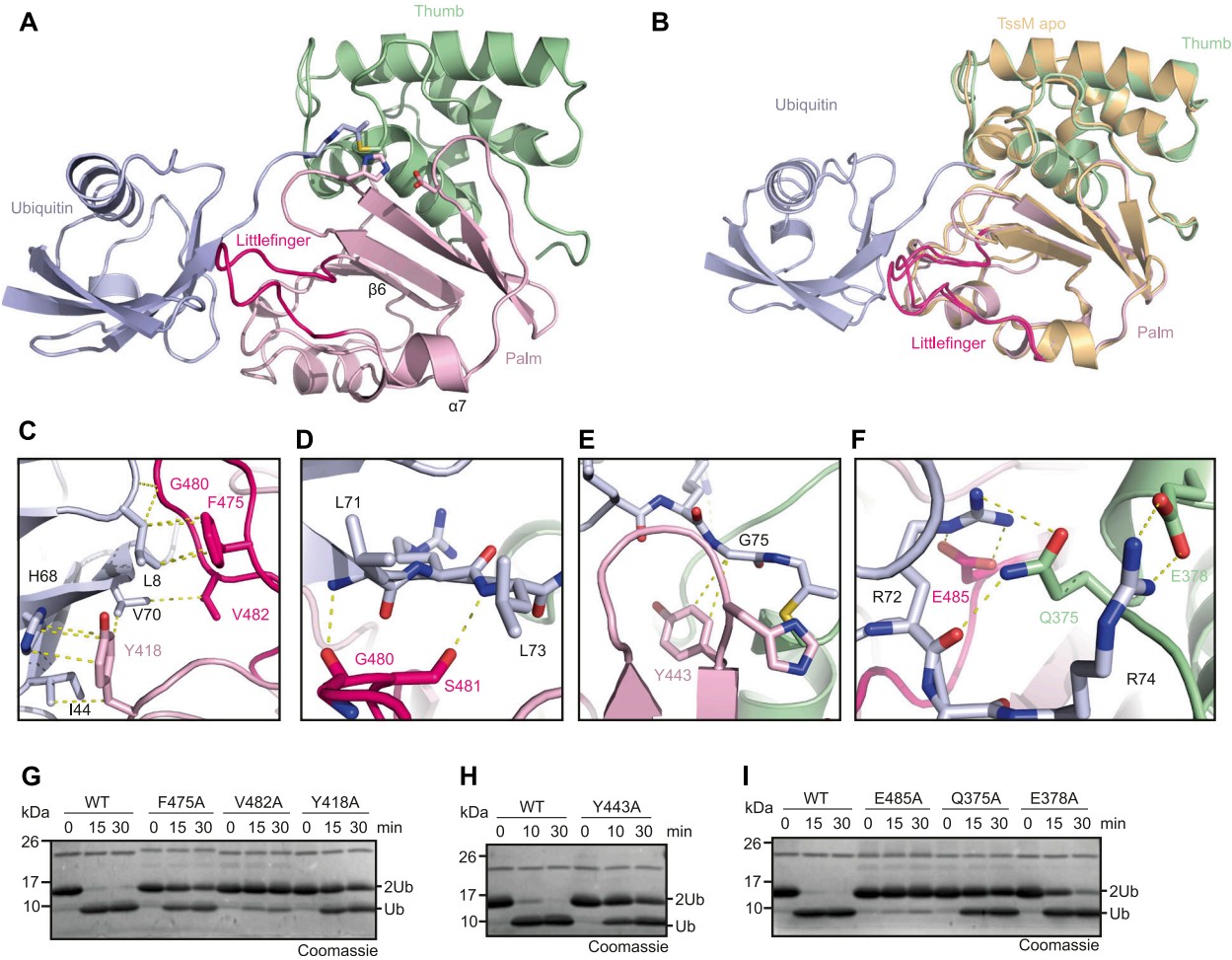

**Figure 3. Novel Littlefinger loop functionally replaces the Fingers subdomain.**
**(A)** Crystal structure of the TssM$^{193–490}$/Ub-PA complex shown in cartoon representation. The catalytic domain is divided in the Thumb and Palm subdomains and colored green and light pink, respectively. A flexible loop located between α-7 and β-6 is located close to ubiquitin (light blue) and named Littlefinger (pink).
**(A, B)** Structural comparison of apo (colored light orange) and Ub-PA-bound (subdomains colored as in panel (A)) TssM. The catalytic domains are shown in cartoon representation and were superimposed with a RMSD of 0.35 Å over 1,115 atoms. **(C)** Recognition of ubiquitin's Ile-44 patch by residues of the Littlefinger loop and the Palm subdomain. Ubiquitin (light blue) and TssM are shown in cartoon representation. The Littlefinger is colored pink and the Palm subdomain is colored light pink. Key residues are shown as sticks and hydrophobic interactions are indicated by dotted lines. **(D)** Littlefinger (colored pink) residues stabilize ubiquitin C-terminus (light blue). Key residues are shown as sticks and hydrogen bonds are indicated by dotted lines. **(E)** Interaction between Gly75 of ubiquitin and the "aromatic motif" represented by Tyr443, shown as light pink stick, is conserved in TssM. Hydrophobic interactions are indicated by dotted lines. **(F)** Recognition of ubiquitins Arg72/Arg74 by TssM. The conserved residue Glu378 forms a salt bridge with Arg74, in contrast to its role in classic USPs, where it contacts Arg72. TssM stabilizes Arg72 instead via Glu485 and Gln375. Key residues are shown as sticks and colored according to the subdomains: Littlefinger/pink, Thumb/green. **(G)** Mutational analysis of residues contacting the Ile-44 patch. 1.5 μM WT TssM$^{292–490}$ and the respective point mutants were incubated with K48-linked di-ubiquitin. The reaction was stopped by the addition of Laemmli buffer after the indicated timepoints, resolved by SDS–PAGE and Coomassie-stained. **(H)** Comparison of WT TssM and the aromatic motif mutant Y443A. 1.5 μM TssM$^{292–490}$ were incubated with K48-linked di-ubiquitin and the reaction was stopped by the addition of Laemmli buffer after the indicated timepoints. **(I)** Mutational analysis of residues stabilizing ubiquitin's C-terminal Arg72/Arg74 residues. 1.5 μM WT TssM$^{292–490}$ and the respective point mutants were incubated with K48-linked di-ubiquitin. The reaction was stopped by the addition of Laemmli buffer after the indicated timepoints, resolved by SDS–PAGE and Coomassie-stained.
Source data are available for this figure.

was not resolved (chain A) or adopted an irregular conformation (chain C) in the complex structure (Fig 4C), suggesting that this helix is only formed during the β-strand swap and might act as a hinge region. Despite their connection via a flexible linker, the orientation of the Ig-like domain relative to the catalytic domain was surprisingly well conserved in both structures (Fig 4D). The domains are locked in this position by a series of inter-domain salt bridges. These are likely the same in both crystals; however, the apo form does not resolve them unambiguously (Fig 4E).

The N-terminal domain of TssM is structurally related to the extensive class of bacterial Ig-like domains, which occur—often in tandem arrangement—in bacterial adhesins, surface layer proteins, and enzymes (Bodelon et al, 2013). As expected for cytoplasmic proteins, the Ig-like domain of TssM is not stabilized by disulfide bridges, which are a hallmark of classical immunoglobulins. Structural comparisons to solved structures in the PDB database using Dali yielded more than 120 structures with Z-scores greater than 6.0, indicating high structural similarity. The best scores were

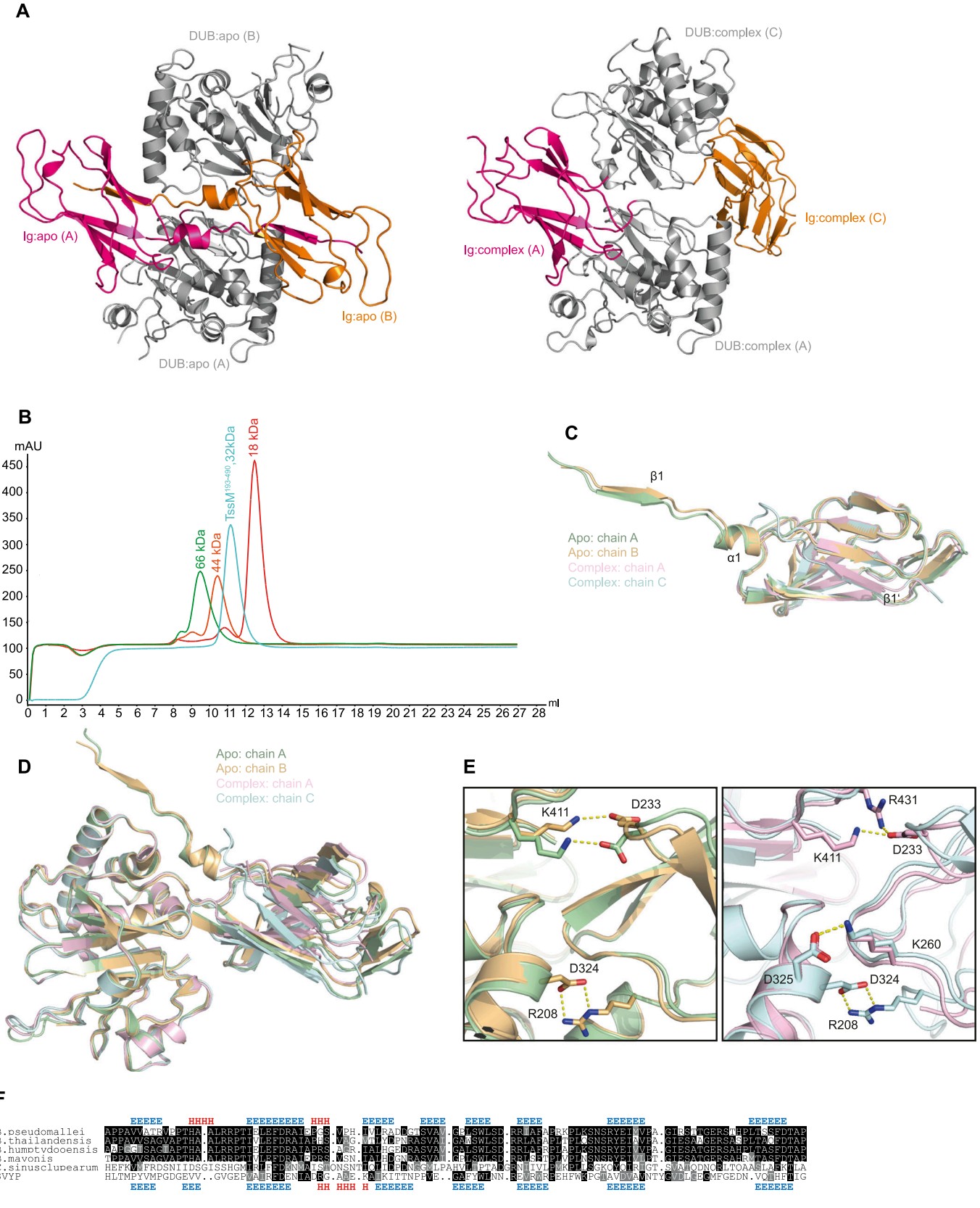

found for surface layer proteins (PDB:4UIC), followed by copper resistance protein C (PDB:6NFR), and L,D-transpeptidases (PDB: 3TUR). To gain insights into the evolutionary origins of the TssM N-terminal domain, sequence similarity is often more informative than structural similarity, because it is less prone to reporting similarities caused by convergent evolution. HMM-to-HMM searches using an alignment of Ig-like domains found in the TssM family revealed highly significant sequence matches with several L,D-transpeptidases. The best score ($1.3 \times 10^{-7}$) was obtained for the sequence family representing pdb:3VYP, a transpeptidase from *Mycobacterium tuberculosis* (Li et al, 2013). A sequence alignment highlighting the similarity of the TssM-Ig domain to the transpeptidase is shown in Fig 4F. In summary, our in vitro data do not pinpoint a clear function of the TssM Ig-like domain. Because it is not important for DUB activity, a role in recruiting TssM to its site of action is likely.

# Discussion

At the outset of this study, we considered TssM to be a divergent member of the USP family with a large internal deletion, affecting a USP-typical subdomain crucial for ubiquitin binding; we also considered a possible relationship with the Josephin family of deubiquitinases. Structural analysis confirmed the complete absence of the Fingers subdomain, which is invariably present in all known USP structures, even in USPL1 (Li et al, 2022), a divergent USP family member that cleaves SUMO rather than ubiquitin (Schulz et al, 2012). CYLD, an even more divergent USP-like deubiquitinase, was reported to contain a "truncated" Fingers region, meaning that the β-strands are shortened and the two β-strands forming the "Fingertip" are absent (Komander et al, 2008). By contrast, TssM does not contain a single β-strand at this position, owing to the internal deletion of a sequence stretch covering the middle of α5 through β8 in the USP7 reference structure. As a replacement, TssM uses a single α-helix flanked by two flexible linkers to connect the Thumb and Palm subdomains. Structurally, this replacement loop forms part of the extended Palm subdomain, which—like the Thumb domain—is otherwise conserved in TssM. However, the relative positioning of the Palm and Thumb subdomains is slightly different from that seen in canonical USPs, which explains the relatively poor RMSD values obtained by superposition of the entire catalytic domain.

USP-type enzymes bind the S1 ubiquitin via their Fingers subdomains, as shown for USP7 (Hu et al, 2002). Most prominently, residues located within the Fingers region form hydrogen bonds with Thr-66 and van der Waals interactions with Phe-4 of ubiquitin (Hu et al, 2002). Because of the absence of the Fingers region, this ubiquitin interface remains entirely untouched in the solved TssM/Ub-PA complex structure. This surface is also not used by Josephin-type DUBs, which bind the S1-ubiquitin mainly via the Ile-44 patch (Weeks et al, 2011). The structure of the covalent TssM:Ub complex also revealed extensive contacts to the Ile-44 patch. However, this binding mode is different from Josephin, because it uses a completely different part of the catalytic domain, and does therefore not support an evolutionary relationship (Fig S3D). TssM uses the loop connecting α7 and β6 to recognize the Ile44 patch. This region, which we refer to as Littlefinger, is functionally, but not structurally, analogous to the Fingers region of classical USPs. The Littlefinger loop and its ubiquitin–recognition properties appear conserved in TssM-like sequences from selected other *Burkholderia* species, including *B. mallei*, *Burkholderia thailandensis*, *Burkholderia mayonis*, and *Burkholderia humptydooensis* (Fig S4A–C). Outside of Burkholderia, two TssM-like DUBs could be identified, one from *Chromobacterium sinusclupearum*, the other one from an unidentified metagenomic sequence. However, these two sequences are predicted to lack the Littlefinger-based ubiquitin recognition (Fig S4A, D, and E).

Interestingly, the Littlefinger region functionally replaces two additional USP-typical binding interfaces. On the one hand, canonical USPs stabilize the ubiquitin C-terminal residues Leu-71 and Leu-73 via a hydrophobic pocket. In TssM, this interaction is replaced by hydrogen bonds between the main chain of these residues and the Littlefinger. On the other hand, canonical USPs possess a highly conserved acidic residue within the Thumb region, which forms a salt bridge with Arg-72 of ubiquitin. In TssM, this task is fulfilled by a residue within the Littlefinger. Interestingly, the canonical acidic residue is also conserved in TssM, but forms a salt bridge with Arg-74 rather than Arg-72. Josephins also form salt bridges with Arg-72/74, but their positioning is completely different, as the bound ubiquitin is rotated by 180° (Fig S3D). Taken together, the ubiquitin binding of TssM differs markedly from eukaryotic USPs, but even more so from the Josephin family. In all eukaryotic DUB families (except ZUFSP), a conserved aromatic residue after the catalytic histidine residue fulfills two separable functions. On the one hand, it acts as a gatekeeper and only allows modifiers with a

**Figure 4. Conserved Ig-like domain.**
**(A)** Arrangement of the TssM molecules in the ASU of the apo (left) and the ubiquitin complex crystals (right). The catalytic domains are colored in grey and the Ig-like domains are colored pink (chain A) and orange (chain B or C). All ubiquitins have been omitted for the sake of the comparison of the TssM molecules. Both ASUs were aligned on their chain A-TssM (bottom). The other TssM (top) is rotated by ~70°. Hence, both Ig-like domains are separated from each other in the complex crystal, although they are in close proximity swapping their first β-strand in the apo one. **(B)** Size exclusion chromatography to determine the dimer formation in solution. TssM$^{193–490}$ (teal), myoglobin (18 kD, red), ovalbumin (44 kD, orange), and bovine serum albumin (66 kD, green) were individually subjected to size exclusion chromatography and the resulting UV curves were merged. **(C)** Structural superposition of the Ig-like domains derived from the TssM apo and complex structure. Each Ig-like domain was superimposed on the chain A Ig-like domain from the apo structure. RMSD: 1.42 (apo A/apo B; 1,330 atoms), 1.33 (apo A/complex A; 563 atoms), and 1.08 Å (Comp C/apo A 522 atoms). Secondary structure elements differing in the structures are numbered. **(D)** Structural superposition of TssM$^{193–490}$ from the apo and complex structure, based on the respective catalytic domains. **(E)** Close-up view on the superposition shown in (E) highlighting the inter-domain salt bridges. The left panel shows TssM from the apo structure and the right panel shows the complex structure. Residues forming salt bridges are shown as sticks and the resulting salt bridges are indicated by yellow dotted lines. **(F)** Structure-guided alignment of TssM-derived Ig-like domains from different species, together with a structurally characterized transpeptidase (PDB:3VYP). Residues printed on black or grey background are invariant or conservatively replaced in at least 50% of the sequences. Secondary structure elements are indicated on top (TssM) or bottom (3VYP) of the alignment by colored letters (E = β-strand, H = α-helix).

glycine at the penultimate position to reach the active site. In addition, its interaction with Gly75 is crucial for catalytic activity, suggesting an additional stabilizing role (Hermanns et al, 2020). At least the latter role also applies to the TssM aromatic motif, as demonstrated in Fig 3H.

TssM is highly selective for ubiquitin (Fig 1B–D); the basis for this specificity becomes apparent from a number of structural contacts. The dependence on the Ile-44 patch recognition prevents cross-reactivity with UbLs lacking this patch, such as ISG15 or SUMO. NEDD8 shares the Ile44 patch with ubiquitin but is a much poorer substrate. This finding can be explained by an important hydrogen bond between the Littlefinger region and Arg-72 of ubiquitin. In NEDD8, Arg-72 is replaced by an alanine residue, thereby preventing this crucial stabilization.

Considering that the major function of bacterial DUBs is the reversal of ubiquitination, which might target bacteria or bacteria-containing vacuoles for autophagic degradation, the potential for Nedd8 cross-reactivity might be of minor importance. Most intra-cellular bacteria studied so far encode one or two DUB effectors, often belonging to the OTU or CE-Clan enzyme families and exhibiting little linkage preference (Pruneda et al, 2016; Hermanns & Hofmann, 2019; Schubert et al, 2020). A remarkable exception is *L. pneumophila*, which codes for more than 10 different deubiquiti-nases, some of which have exquisite linkage specificity (Wan et al, 2019; Hermanns et al, 2020; Shin et al, 2020; Warren et al, 2023). For *B. pseudomallei* and *B. mallei*, TssM is the only DUB effector described to date, and our bioinformatics searches in the respective genome sequences did not yield additional DUB candidates. Compared with other known DUB effectors, TssM is unusual in two respects. First, it is the first and only bacterial USP-type enzyme to be described. Moreover, TssM has been shown to be secreted into the host cy-toplasm by a type-II secretion system, whereas other bacterial DUB effectors depend on type-III, type-IV or type-VI secretion system (Burtnick et al, 2014). Type-II secretion systems transport bacterial translation products across the inner and outer bacterial mem-branes, but unlike T3SS, T4SS and T6SS do not support secretion through an additional host-derived membrane. This secretion mode implies that Burkholderia effectively secretes TssM only after entering the host cell and escaping the phagosome (Allwood et al, 2011). Therefore, the expected function of TssM is the removal of ubiquitin conjugates from the bacterial surface rather than from bacteria-containing vacuoles. A number of mammalian ubiquitin ligases have been proposed to modify bacterial surfaces, most prominent among them is RNF213, which directly ubiquitinates LPS (Otten et al, 2021). Other ligases can further modify the initial ubiquitin by various chain types, such as the linear chains installed by LUBAC onto escaped Salmonella bacteria (Noad et al, 2017) or the K6-linked chains generated by LRSAM1 (Huett et al, 2012). Because TssM does not cleave linear chains and is poorly active against K6 chains, a role for TssM in removing the first ubiquitin from the bacterial surface appears most plausible.

The non-catalytic immunoglobulin-like domain of TssM is found in all TssM-like enzymes (Fig 4F) and might be involved in targeting TssM to its substrate. The closest sequence relatives of the TssM Ig-like family are Ig-like domains of LD transpeptidases, which are also non-catalytic and precede the transpeptidase domain (Both et al, 2013); however, their function is not well understood. Because LD

transpeptidases act as peptidoglycan crosslinkers, the non-catalytic portion of these enzymes might play a role in sugar binding. Alternatively, a spacer function has been proposed, which would allow the plasma membrane-bound transpeptidases to reach their substrates within the periplasmic space (Both et al, 2013). Similar functions have been proposed for other bacterial Ig-like domains (Bodelon et al, 2013). The positioning of the Ig-like domain between an extended hydrophobic region and the catalytic domain suggests that the TssM N-terminal domain helps to direct the deubiquitinase activity to the bacterial surface. Alternatively, the detachable first β-strand of the Ig-like domain may assist type-II secretion, which has been proposed to involve a β-strand complementation mechanism (Korotkov et al, 2012). The β-strand swap observed in the apo-structure (Fig 4A) does not appear to stabilize a TssM dimer in solution and might be a crystallization artifact (Fig 4B). However, a dimerization under different conditions, for example, on the surface of bacteria, cannot be excluded. Al-though the positional fixation of the Ig-like domain relative to the USP domain by multiple salt bridges (Fig 4D and E) might imply a functional coupling of the two domains, this arrangement is clearly dispensable for catalysis (Fig 2E and F). For a better understanding of the contributions made by the Ig-like domain, an infection model will have to be used.

Recently, an independent report on the structure of TssM and its catalytic properties was published as a preprint (Szczesna et al, 2023 *Preprint*). This work focusses on the functional implications of TssM to help *Burkholderia* evade RNF213-mediated surface ubi-quitination. The structure of the TssM ~ Ub complex provided by Szczesna et al is very similar to the complex structure described here; the same is true for the identified ubiquitin recognition determinants and their influence on TssM catalytic activity.

# Materials and Methods

## Sequence analysis

All sequence alignments were generated using the MAFFT package (Katoh & Standley, 2013). Generalized profiles were calculated from multiple alignments using pftools (Bucher et al, 1996), and searched against all proteins from the Uniprot database (https://www.uniprot.org) and the NCBI microbial genome reference se-quence database (https://www.ncbi.nlm.nih.gov/genome/microbes). HMM-to-HMM searches were performed using the HHSEARCH method (Soding, 2005), searches against precomputed PDB database was performed using the HHpred toolkit (Zimmermann et al, 2018). Structure predictions were performed using a local installation of Alphafold 2.1 (Jumper et al, 2021). For structure comparisons, the DALI software was used (Holm, 2020).

## Cloning and mutagenesis

TssM[193–490] coding region was obtained by gene synthesis (IDT) and cloned into pOPIN-K vector (Berrow et al, 2007) using the In-Fusion HD Cloning Kit (Takara Clontech). TssM[292-490] was amplified from pOPIN-K-TssM[193–490] and cloned into pOPIN-S vector (Berrow et al,

2007) using the In-Fusion HD Cloning Kit (Takara Clontech). Point mutations were introduced using the QuikChange Lightning kit (Agilent Technologies).

Constructs for ubiquitin–PA purification (pTXB1-ubiquitin[1–75]) were a kind gift of David Komander (WEHI, Melbourne). SUMO1[1–96] was amplified by PCR with an N-terminal 3xFlag tag, SUMO2[1–92] and ISG15[79–156] without tag and subsequently cloned into the pTXB1 vector (New England Biolabs) by restriction cloning according to the manufacturer's protocol.

## Protein expression and purification

TssM[193–490] was expressed from the pOPIN-K vector with an N-terminal 6His-GST-tag, whereas TssM[292–490] and all point mutants were expressed from pOPIN-S vector with an N-terminal 6His-Smt3-tag. *Escherichia coli* (Strain: Rosetta [DE3] pLysS) were transformed with constructs expressing DUBs and 2–6l cultures were grown in LB medium at 37°C until the $OD_{600}$ of 0.8 was reached. The cultures were cooled down to 18°C and protein expression was induced by addition of 0.1 mM isopropyl $\beta$-d-1-thiogalactopyranoside (IPTG).

After 16 h, the cultures were harvested by centrifugation at 5,000$g$ for 15 min. After freeze–thaw, the pellets were resuspended in binding buffer (300 mM NaCl, 20 mM TRIS pH 7.5, 20 mM imidazole, 2 mM $\beta$-mercaptoethanol) containing DNase and lysozyme, and lysed by sonication using 10-s pulses with 50 W for a total time of 10 min. Lysates were clarified by centrifugation at 50,000$g$ for 1 h at 4°C and the supernatant was used for affinity purification on HisTrap FF columns (GE Healthcare) according to the manufacturer's instructions. The 6His-Smt3 tag was removed by incubation with SENP1[415–644]; the 6His-GST tag was removed by incubation with 3C protease. The proteins were simultaneously dialyzed in binding buffer. The liberated affinity-tag and the His-tagged SENP1 and 3C proteases were removed by a second round of affinity purification with HisTrap FF columns (GE Healthcare). All proteins were purified with a final size exclusion chromatography (HiLoad 16/600 Superdex 75 pg) in 20 mM TRIS pH 7.5, 150 mM NaCl, 2 mM DTT, concentrated using VIVASPIN 20 Columns (Sartorius), flash frozen in liquid nitrogen, and stored at –80°C. Protein concentrations were determined using the absorption at 280 nm ($A_{280}$) using the proteins' extinction coefficients derived from their sequences.

## Synthesis of activity-based probes

All activity-based probes used in this study were expressed as C-terminal intein fusion proteins. The intein fusion proteins were affinity purified in buffer A (20 mM HEPES, 50 mM sodium acetate pH 6.5, 75 mM NaCl) from clarified lysates using Chitin Resin (New England Biolabs) following the manufacturer's protocol. On-bead cleavage was performed by incubation with cleavage buffer (buffer A containing 100 mM MesNa [sodium 2-mercaptoethanesulfonate]) for 24 h at RT. The resin was washed extensively with buffer A and the pooled fractions were concentrated and subjected to size exclusion chromatography (HiLoad 16/600 Superdex 75 pg) with buffer A. To synthesize the propargylated probe, 300 $\mu$M Ub/Ubl-MesNa were reacted with 600 mM propargylamine hydrochloride (Sigma-Aldrich) in buffer A containing 150 mM NaOH for 3 h at RT. Unreacted propargylamine was removed by size exclusion chromatography and the probes were

concentrated using VIVASPIN 20 Columns (3 kD cutoff; Sartorius), flash frozen, and stored at –80°C. The NEDD8-PA was a kind gift from David A Pérez Berrocal and Monique PC Mulder (Department of Cell and Chemical Biology, Leiden University) (Ekkebus et al, 2013; Perez Berrocal et al, 2023).

## Chain generation

Met1-linked di-ubiquitin was expressed as a linear fusion protein and purified by ion exchange chromatography and size exclusion chromatography. K11-, K48-, and K63-linked ubiquitin chains were enzymatically assembled using UBE2S$\Delta$C (K11), CDC34 (K48), and Ubc13/UBE2V1 (K63) as previously described (Komander & Barford, 2008; Bremm et al, 2010). In brief, ubiquitin chains were generated by incubation of 1 $\mu$M E1, 25 $\mu$M of the respective E2, and 2 mM ubiquitin in reaction buffer (10 mM ATP, 40 mM TRIS [pH 7.5], 10 mM $MgCl_2$, 1 mM DTT) for 18 h at RT. The respective reactions were stopped by 20-fold dilution in 50 mM sodium acetate (pH 4.5) and chains of different lengths were separated by cation exchange using a Resource S column (GE Healthcare). Elution of different chain lengths was achieved with a gradient from 0 to 600 mM NaCl.

## Crystallization

100 $\mu$M TssM[193–490] were incubated with 200 $\mu$M ubiquitin-PA for 18 h at 4°C. Unreacted TssM[193–490] and Ub-PA were removed by size exclusion chromatography. TssM[193–490] alone and the covalent TssM[193–490]/Ub-PA complex (10 mg/ml) were crystallized using the vapor diffusion with commercially available sparse matrix screens. Crystallization trials were set up with drop ratios of 1:2, 1:1, 2:1 protein solution to precipitant solution with a total volume of 300 nl.

Initial crystals of apo TssM[193–490] appeared in JCSG A9 (0.2 M ammonium chloride, 20% wt/vol PEG3350) at 20°C. These crystals were optimized by addition of 0.3 $\mu$l of different additives to 3 $\mu$l protein/precipitant drops (Additive Screen; Hampton Research). Best diffracting crystals were harvested from a condition containing 0.15 M ammonium chloride, 22% wt/vol PEG 3350, and 4% dextran sulfate sodium salt Mr 5,000 and were cryoprotected with a reservoir solution containing 25% glycerol.

Initial crystals of TssM[193–490]/Ub-PA appeared in Peg/Ion D12 (0.2 M ammonium citrate dibasic, 20% wt/vol PEG3350) at 20°C. These crystals were optimized by gradually changing the ammonium citrate and PEG3350 concentrations using 48-well MRC plates with 80 $\mu$l reservoir solutions and 3 $\mu$l drops (protein/precipitant ratios: 2:1, 1:1, and 1:2). Best diffracting crystals were harvested from a condition containing 0.2 M ammonium citrate dibasic and 22% wt/vol PEG 3350 and were cryoprotected with a reservoir solution containing 25% ethylenglycol.

## Data collection, phasing, model building, and refinement

Diffraction data of a TssM[193–490] apo-crystal were collected at the European Synchrotron Radiation Facility, France at beamline ID30B at 0.92 Å wavelength. Diffraction data of a TssM[193–490]/Ub-PA complex crystal were collected at the Swiss Light Source, Switzerland, at beamline X06SA at 1 Å wavelength. The datasets were

processed using XDS (Kabsch, 2010) and encompass reflections up to 3.15 and 1.62 Å (CC1/2 of 0.5), respectively. An AlphaFold (Jumper et al, 2021) prediction was split into the Ig-like and the catalytic domains and both were used individually as molecular replacement search models. The complex was solved using 1UBQ (Vijay-Kumar et al, 1987) for ubiquitin and the individual domains from the almost-final apo-TssM model. The first $\beta$-strand which is swapped in the apo-model had been removed before hand. Molecular replacement was carried out using PHASER (McCoy et al, 2007) as implemented in the phenix package (Adams et al, 2010). Initial models were refined using iterative cycles of phenix.refine and manually rebuilt using COOT (Emsley et al, 2010). A final refined with Refmac (Murshudov et al, 1997) was done for the TssM$^{193-490}$/Ub-PA complex. For structural analysis, the PyMOL (http://www.pymol.org) and ChimeraX Graphics Systems (Pettersen et al, 2021) were used.

### AMC assays

Activity assays of DUBs against AMC-labeled substrates were performed using reaction buffer (150 mM NaCl, 20 mM TRIS pH 7.5, 10 mM DTT), 5 $\mu$M Ub-AMC (UbiQ-Bio) or 5 $\mu$M Nedd8-AMC (Enzo Life Science). The TssM concentrations are stated in the respective figure legends. The reaction was performed in black 96-well plates (Corning) at 30°C and fluorescence was measured using the Infinite F200 Pro plate reader (Tecan) equipped for excitation wavelength of 360 nm and an emission wavelength of 465 nm. The presented results are means of three independent cleavage assays. The initial velocities of ubiquitin and NEDD8 cleavage were determined using the linear range of 5 nM TssM versus 5 $\mu$M Ub/NEDD8-AMC. The measurements were performed in triplicates and the presented results are the mean and the SD of the individual measurements.

### Activity-based probe assays

DUBs were prediluted to 2× concentration (10 $\mu$M) in reaction buffer (20 mM TRIS pH 7.5, 150 mM NaCl and 10 mM DTT) and combined 1:1 with 100 $\mu$M Ub-PA, NEDD8-PA, ISG15$^{CTD}$-PA, FLAG-SUMO1-PA or SUMO2-PA for 18 h at 20°C. The reaction was stopped by the addition of 2x Laemmli buffer, and analyzed by SDS–PAGE using Coomassie staining.

### Ubiquitin chain cleavage

DUBs were prediluted in 150 mM NaCl, 20 mM TRIS pH 7.5 and 10 mM DTT. The cleavage was performed at 20°C for the indicated time points with different TssM concentrations (as indicated in the respective figure legends) and 25 $\mu$M di-ubiquitin (M1, K11, K48, K63 synthesized as described above, K6, K29, K33 purchased from Biomol, K27 from UbiQ). The reactions were stopped with 2x Laemmli buffer, resolved by SDS–PAGE, and Coomassie stained.

### Analytical size exclusion chromatography

TssM$^{193-490}$ and the calibration proteins (myoglobin 17.8 kD, ovalbumin 44 kD, and bovine serum albumin 66 kD [Merck]) were prediluted to 8 $\mu$g/$\mu$l in SEC buffer (20 mM TRIS pH 7.5, 150 mM NaCl, 2 mM DTT). 250 $\mu$l of protein solution were loaded onto a Superdex

75 10/300 GL gel filtration column (GE Healthcare) with a flow rate of 0.5 ml/min. The resulting UV-curves were overlaid using Unicorn 7.0.

## Data Availability

The X-ray structures of TssM and the TssM/Ub-PA complex have been deposited at the PDB database under the accession numbers 8PZ3 and 8Q00, respectively. Source data underlying the findings of this study are provided with this article.

## Supplementary Information

## Acknowledgements

We thank Christiane Horst for expert technical assistance. We thank David A Pérez Berrocal and Monique PC Mulder for the kind gift of Nedd8-PA activity-based probe. The synchrotron data were collected at the Swiss Light Source, Paul-Scherrer-Institute, Switzerland, at beamline X06SA and at the European Synchrotron Radiation Facility (ESRF), France at beamline ID30B. We thank the staff for all support and the maintenance of the accurate machines. Deubiquitinase research in the laboratory of K Hofmann is supported by DFG Grant HO 3783/3-1. Crystals were grown using equipment of the Cologne Crystallization facility (C$_2$f), which is supported by DFG Grant INST 216/949-1 FUGG. We acknowledge support for the Article Processing Charge from the DFG (German Research Foundation, 491454339).

### Author Contributions

T Hermanns: conceptualization, supervision, investigation, visualization, and writing—original draft, review, and editing.
M Uthoff: investigation, and writing—original draft, review, and editing.
U Baumann: supervision, funding acquisition, investigation, and writing—original draft, review, and editing.
K Hofmann: funding acquisition, investigation, visualization, and writing—original draft, review, and editing.

### Conflict of Interest Statement

The authors declare that they have no conflict of interest.

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
