## [Reviewer comments · Life Science Alliance]

Life Science Alliance

The structural basis for deubiquitination by the fingerless USP-type effector TssM

Thomas Hermanns, Matthias Uthoff, Ulrich Baumann, and Kay Hofmann

DOI: <https://doi.org/10.26508/lsa.202302422>

Corresponding author(s): Thomas Hermanns, University of Cologne

Review Timeline:

Submission Date:	2023-10-06
Editorial Decision:	2023-11-07
Revision Received:	2023-11-24
Editorial Decision:	2023-11-27
Revision Received:	2023-11-28
Accepted:	2023-11-29

Transaction Report:

Please note that the manuscript was reviewed at *Review Commons* and these reports were taken into account in the decision-making process at *Life Science Alliance*.

Review #1

Hermanns et al. report a robust biochemical and structural characterization of the TSSM virulence effector of *Burkholderia pseudomallei*, the causative agent of melioidosis, with TSSM being the only USP-type deubiquitinase of bacterial origin. The authors demonstrate ubiquitin specificity, and the capacity of the DUB to cleave most Ubiquitin chain types *in vitro*. By solving crystal structures of TSSM in isolation as well as in covalent complex with Ubiquitin, the authors rationalize how the truncated fold (compared to all other eukaryotic USPs) is capable of binding ubiquitin. The most surprising finding is the conformation of the so-called little finger loop, unique to TSSM, which engages the distal Ubiquitin in a manner not seen in any other DUB. These findings are convincingly validated by point mutations in biochemical assays and bioinformatic analyses. The structures also yielded information on the fold an N-terminal Ig-like domain with unknown function.

The authors are encouraged to address the following main points before publication can be supported:

- On page 6, the authors state "the little finger loop contacts hydrophobic residues around Ile-44 of ubiquitin, an important interaction interface that is not observed in USP7 and is not part of the canonical USP: ubiquitin interface". The latter statement I believe is simply wrong as Ile44 is at the centre of the large canonical USP:ubiquitin interaction interface observed in all structurally studied USP proteins so far (the recognition is through varying residues, that are not always strictly hydrophobic, but it is always contacted). This is occurring at the hinge between thumb and fingers. In USP7, this is for example occurring through the hydrophobic part of Arg301. The authors are referred to O'Dea et al. Nat Chem Biol 2023, where same conformation of the equivalent residue in USP36 is discussed in the context of Ile44-Ubiquitin vs. Fubi recognition.
- This should also be phrased very precisely, as Ile44 itself is not actually contacted by the little finger loop (see Fig. 3c), but other hydrophobic residues commonly included in the Ile44 patch are. However, also those are always in the Ubiquitin:USP interface.
- It appears as if the little finger loop takes up the exact position of what is commonly referred to as blocking loop 1 (terminology introduced by Hu et al EMBO J 2005, which the authors could also use when referring to canonical USPs). The connectivity in the domain is not equivalent to BL1 though (this could be shown with topology diagrams of a canonical USP and of TSM), but the relative position is. Importantly, BL1 in USPs typically engage the Ile36 patch with a large aromatic residue (e.g., a Trp in USP30 or a Tyr in USP7), however, in TSSM Ubiquitin is rotated such that the Ile44 patch is recognised at the exact spot where normally Ile36 is engaged. Can the authors comment on this interpretation? This reviewer would also find a comparison to the canonical USP recognition mode in the main figures beneficial (however one that is less crowded than in the SI) to make this very important finding come across well. Even if the loop is more like a BL1 (which is highly diverse in the USP family, see reference above), it can still be called "little finger", but the analogies should be carefully phrased throughout the text (e.g., page 9: "functionally but not structurally analogous" would not be correct if it does not replace the fingers which contact Phe4 as BL1 is not part of the fingers).
- The assessment of the molecular weight of apo TSSM in solution by SEC is flawed in its current form (page 7, Fig. 4b). There are only two, seemingly randomly chosen, comparison proteins, and the GST-Ub is unsuitable for 36 kDa because it is an obligate dimer (due to the GST) and as a fusion protein has a much higher hydrodynamic radius than a globular protein. There are commercial reference proteins that can be used which have defined oligomerisation states and are (reasonably globular) so that the retention times can be quantitatively analysed. Such a reference, which also includes proteins of smaller sizes, must be used. Alternatively, the authors should use light-scattering (SEC-MALS or SEC-RALS) to unequivocally demonstrate the molecular weight / oligomeric state of TSSM.
- The abstract ends with speculation on the presence of the strand-swapping in live cells and of the role of the Ig-fold domain, but not a proper conclusion. The same applies to the ending of the results section (page 8), where it is not clear where the structural analysis of the Ig-like domains is leading. The authors then speculate about sugar binding (page 10) in the discussion. The latter could be substantiated by an analysis of the residues in the possible sugar binding site (if present in their TSSM), and the text be more rounded off at these regions.
- The authors show in Fig. 1b TSSM reactivity with both Ubiquitin and Nedd8 probes but then qualify the Nedd8-reactivity with the assays shown in Figs. 1cd, which currently looks like it is an issue with the comparison of probe to substrate. However, their probe assay is very long with 18 h. It should be repeated at shorter times (for Ub and Nedd8-PA only) to test if the strong Ub preference seen in the kinetic assay can also be visualized with the probes.
- Moreover, the authors prepare Ubl-PA probes by aminolysis of C-terminal thioesters but at very different

stoichiometries as reported for Ub-PA (e.g., in Gersch et al. NSMB 2017). The precise amounts should be cross-checked. This reviewer would not be surprised if the conditions of only 2-fold excess of propargylamine over Ubl thioester and the high amount of NaOH led also to hydrolysis of the C-terminus, which by size exclusion chromatography would not be separated. For all probes, intact mass spectra of the used aliquots must be shown to demonstrate the identity of the used reagents. The presence of other species of course does not per se disqualify the assay in Fig. 1b.

In addition, the following minor points should be addressed:

- On page 2, the authors state that the 1-2 DUBs typically present in bacteria either target K63-chains or lack specificity (without reference), but on page 10 they state that bacteria typically only have 1 DUB with little specificity. This is contradicting and should be fixed.
- On page 3, the authors make the case for TSSM from *B. pseudomallei* by calling it representative. This reviewer would appreciate if the authors could expand on this towards the end of the manuscript and comment on whether they expect the identified Ubiquitin recognition mode to be present also in all (!) other DUB-TSSM proteins, at least all analysed bioinformatically. Importantly, they should include the sequence of TSSM of *B. mallei* (the only in vitro studied TSSM so far) into their analyses in the SI.
- On page 4, the authors start with TSSM (193-490) but do not comment on the role of the first 192 residues. What was the rationale for these boundaries?
- On page 4, the authors introduce the use of the fluorogenic AMC substrates with "more quantitative analysis", however, the experiments are rather minimalistic. Only one substrate concentration, and two enzyme concentrations (at least for one substrate) are used, and the data are not really analysed in a quantitative manner. Through curve fitting of the existing data (with fixed restraints) the authors may be able to determine an estimate of the Ub/Nedd8-specificity factor. Moreover, negative controls should be shown in Figure 1c to demonstrate that the Nedd8-curve is above a possible baseline drift.
- On page 4, the authors mention a possible relationship with the Josephin family (which they later disprove through structural comparisons, page 6). It may be helpful to briefly explain the rationale (i.e., why one would even consider a relationship with the Josephin family DUB given the higher homology to the USP fold).
- On page 5, the "shortened C-terminus" compared to USP7 is mentioned. This is misleading, as USP7 has an elongated C-terminus compared to most other USPs, and so the TSSM C-terminus is the canonical ending of the USP fold.
- On page 6, "ubiquitin has multiple specific contacts" - why are the contacts named "specific"?
- On page 10, the TSSM from *C. sinucluepearum* appears without any context. Chromobacterium should be spelled out, and it should be discussed if this is the odd one out. It would also be appreciated if the authors could state whether their bioinformatic analysis is comprehensive (i.e., do only the known or all Burkholderia strains have TSSM with a DUB profile). And why is there a A0A1J5... sequence included in Supp Fig. 3a without any context?
- Some minor polishing in the methods would increase consistency (cloning is only mentioned for pOPIN-K, but TSSM is also expressed from pOPIN-S; mutants are only mentioned for the 292-490 construct, but Fig. 2d shows one in the 193-490 construct; Hampton Additive Screens I-III are likely an internal name and not used by Hampton itself; "different TSSM concentrations (as indicated in the figure legends" are mentioned in the methods, but e.g. in the caption to Fig. 1 no concentration is given).
- Likewise, Table 1 needs polishing as commas and points are used interchangeably.
- In Fig. 2c, it looks like the catalytic His is built such that there is no hydrogen bonding between Cys and Asp - is there a particular reason for this? If not, the side chain should be flipped so that the nitrogens are positioned for ideal hydrogen bonding.
- In Fig. 3, dotted lines are introduced as "hydrophobic interactions" as per the captions, but some are clearly hydrogen bonds (e.g., from the amide backbone), and for some others one does not see as they emerge from a cartoon.
- In Fig. 4, context should be given to "3VYP" and why it is used here.

****Referees cross-commenting****

Reviewer comments appear to be fair, balanced and complementary.

Re Reviewer 2's comment on the pre-print: It includes a structure of TSSM bound to ubiquitin (but not of the apo protein). I am not sure if it is appropriate to follow up on the esterase activity. Firstly, there are no tools for it commercially available or readily made in vitro, and secondly it would appear a bit "copycat"-like. Especially since the molecular determinants of what makes a DUB a good esterase are still elusive. A narrower focus of

this manuscript, but done very well (also according to what reviewer 3 suggested), might be a more fitting option.

The findings are novel and of high relevance to the broad ubiquitin and bacterial pathogen communities as the study addresses an enigmatic USP-type deubiquitinase which, as the authors reveal, recognizes Ubiquitin in an entirely different way than its eukaryotic counterparts. This is basic research of pronounced conceptual and mechanistic advance, as it demonstrates that the USP domain can be much more diversified than previously assumed. The structural analysis is very thorough, the data are scholarly presented, and interactions/mechanisms carefully validated by mutations.

The strongest point is clearly the structural analysis of Ubiquitin recognition by this extremely truncated USP fold, and the introduction of the little finger motif. The manuscript does not provide cellular validation of the findings, which is fine to this reviewer for the DUB catalysis part. For the part of the N-terminal domain, the manuscript would benefit from a cellular validation or some localisation studies, however, this is beyond what is established in the authors' lab, and therefore has not been asked. This in turn limits the study to a very thorough in-vitro analysis of this DUB for TSSMs with DUB activity, using conventional substrates like polyUb chains and fluorogenic substrates, and providing convincing conclusions.

My expertise lies in DUBs, biochemistry, and structural biology, and I this believe to have sufficient expertise to evaluate the majority of the findings except the bioinformatic algorithms used which are however not an emphasis in this manuscript.

Review #2

In this manuscript, the authors solved crystal structures of apo-TssM and its complex with Ubiquitin. Together with the biochemical assays, the authors highlighted the differences of TssM from other USP family. TssM do not contain the classical finger domain while it has little finger domain that authors defined. The Ile44 patch on the ubiquitin is mainly used for interacting with TssM.

For the clarity of there findings several points should be edited.

1. In Fig 2d, the authors used different constructs for testing catalytic residues. It will be better to be consistent. Though the authors showed that the deletion of Ig-like domain does not affect the catalytic activity of TssM by showing the AMC assay, it does not guarantee that the effect of mutation on catalytic sites is same for both construct.
2. In fig 2d, authors need to put the label to indicate which linkage of di-Ub is used. Authors did it for figure 2f.
3. By showing AMC assay (fig 2e) and K63 Ub2 cleavage assay (fig 2f), authors concluded that the deletion of Ig-like domain does not affect the activity of TssM. However, as authors found that the Ig-like domain forms dimer at least in their crystallization condition, one cannot exclude the possibility of the role of Ig-like domain in recognising different ubiquitin chains. I would clarify the words by saying "The Ig-like domain does not affect the cleavage K63-Ub2), or authors can expand the cleavage assay with all the linkages.
4. In the apo structure, authors found a domain swapped dimer. One can expect that this dimer is crystallographic artifact and not found in the nature. Indeed, authors could not observe this dimer in the solution when they performed SEC analysis and there was no effect on the catalytic activity when the Ig-like domain is deleted. Because there is no clear evidence of functional importance of this dimer in the manuscript, authors need to clarify about this dimer.

****Referees cross-commenting****

Agree with Reviewer #1's comment on my points.

The structure of TssM is recently reported in a preprint (<https://doi.org/10.21203/rs.3.rs-2986327/v1>) from Pruneda (OHSU). In this preprint, they suggested the role of TssM as Ubiquitin esterase which is not explored in the this manuscript. As it is already published and freely available, authors can explore the role of TssM in that direction as well.

Because the preprint do not contain the complex structure of TssM with Ubiquitin, authors can also examine the roles of Ile44 patch-interacting residues on the catalytic activities.

Also, authors can compare their little finger structure with the preprint.

In general, this manuscript is providing several interesting points to the readers working on the ubiquitin, structures of proteins, host-pathogen interactions and especially those who studying deubiquitinases.

Review #3

In this study, Hermanns et al. have examined the Burkholderia TssM deubiquitinase (DUB) for its ubiquitin chain cleavage specificity using in vitro analyses and for a structural rationalization of its specificity by solving crystal structures with and without covalently bound ubiquitin. TssM was previously shown to be a DUB of the USP family capable of cleaving several ubiquitin chain types. From bioinformatic comparisons, the authors inferred that TssM lacks the 'fingers' domain of classic USP enzymes, and this was shown in the co-crystal structure to be replaced by a 'Littlefinger' loop. The work here is well described and appears to be overall technically solid.

My enthusiasm is reduced for several reasons. First, there is no analysis in an (infected) cell model to evaluate the significance of the TssM mutations, for example, the mutations in ubiquitin-interacting surfaces that are not seen in classic USPs. Second, there is very little quantitation of cleavage rates; while I would not demand derivation of kinetic values for all mutants, the qualitative treatment was not always convincing. For example, Y443A was said to cause "strongly reduced cleavage" (Fig. 4h) whereas E378A was said to "only mildly affect cleavage" (Fig. 4i): to my eye, these cleavage rates are only slightly different (2-3 fold?). Finally, there is a preprint available on Research Square (doi: 10.21203/rs.3.rs-2986327/v1) that shows a potent esterase activity of TssM against ubiquitinated LPS, which is probably its key role in avoiding surveillance and elimination by the host. This paper, although still a preprint, is far more quantitative, includes similar crystallographic data as in the current paper, and describes cellular assays of function. Even without the RS preprint, the Hermanns et al. study provides a fairly modest advance in our knowledge of TssM function; with the preprint, its novelty is, unfortunately, severely reduced.

****Minor comments:****

Full genus names should be spelled out when they first appear in the text (such as Burkholderia and Chromobacterium).

Fig. 3a is described out of sequence.

In Fig 4a, there still seem to be extensive contacts between monomers but the viewing angle could be misleading.

****Referees cross-commenting****

I also agree with Reviewer #1's comments

Even without the Research Square preprint (doi: 10.21203/rs.3.rs-2986327/v1), the Hermanns et al. study provides a fairly modest advance in our knowledge of TssM function; but with the preprint, its novelty is, unfortunately, severely reduced.

1. General Statements

Dear Editor,

We are delighted to see that the three reviewer reports were mostly positive and their demands on further experiments and manuscript changes could easily be dealt with. A detailed description of our changes is provided in the following paragraphs. There are two more general issues, which we would like to briefly clarify here:

1) Two of the reviewers saw a problem (at least with regard to significance) in a recently published preprint by Szczesna et al (<https://www.researchsquare.com/article/rs-2986327/v1>) which has considerable overlap with our manuscript. One reviewer (#2) suggested that we recapitulate the experiments of this preprint, another reviewer (#3) argued that this preprint severely reduces the significance of our own submission. We would like to point out that this Szczesna et al preprint appeared less than two weeks before ours. Actually, we were in contact with the competing group and had planned on a simultaneous submission, which unfortunately got slightly out of sync because we suffered a delay in PDB deposition. The submission process of the Szczesna et al manuscript is at a similar stage as ours. Therefore, we would find it inappropriate to use their data in our work. Moreover, while there is a substantial overlap between the two studies, there are also clear differences: The Szczesna et al preprint focuses on how *Burkholderia* uses TssM to evade RNF213 activity; their work includes infection models and other aspects not covered in our manuscript. Conversely, our focus was on understanding how a divergent USP-type DUB like TssM can be ubiquitin-specific despite the absence of the crucial 'fingers' region. Unlike Szczesna et al, we have solved the TssM structure both in the presence and absence of ubiquitin and can therefore address the structural details of substrate recognition.

2) On a related note, reviewer #3 criticizes that we did not experimentally address the role of TssM in *Burkholderia* infection. We agree that this would have been an interesting study system, but it was clearly not in the focus of our study.

Reviewer #1 (Evidence, reproducibility and clarity (Required)):

Hermanns et al. report a robust biochemical and structural characterization of the TSSM virulence effector of *Burkholderia pseudomallei*, the causative agent of melioidosis, with TSSM being the only USP-type deubiquitinase of bacterial origin. The authors demonstrate ubiquitin specificity, and the capacity of the DUB to cleave most Ubiquitin chain types in vitro. By solving crystal structures of TSSM in isolation as well as in covalent complex with Ubiquitin, the authors rationalize how the truncated fold (compared to all other eukaryotic USPs) is capable of binding ubiquitin. The most surprising finding is the conformation of the so-called little finger loop, unique to TSSM, which engages the distal Ubiquitin in a manner not seen in any other DUB. These findings are convincingly validated by point mutations in biochemical assays and bioinformatic analyses. The structures also yielded information on the fold an N-terminal Ig-like domain with unknown function.

The authors are encouraged to address the following main points before publication can be supported:

- On page 6, the authors state "the little finger loop contacts hydrophobic residues around Ile-44 of ubiquitin, an important interaction interface that is not observed in USP7 and is not part of the canonical USP: ubiquitin interface". The latter statement I believe is simply wrong as Ile44 is at the centre of the large canonical USP:ubiquitin interaction interface observed in all structurally studied USP proteins so far (the recognition is through varying residues, that are not always strictly hydrophobic, but it is always contacted). This is occurring at the hinge between thumb and fingers. In USP7, this is for example occurring through the hydrophobic part of Arg301. The authors are referred to O'Dea et al. Nat Chem Biol 2023, where same conformation of the equivalent residue in USP36 is discussed in the context of Ile44-Ubiquitin vs. Fubi recognition.

We thank reviewer #1 for this detailed explanation. Initially we made this statement based on the fact that in other USPs, the interaction is not always hydrophobic. However, we now agree with reviewer #1 and have removed this statement from the manuscript.

- This should also be phrased very precisely, as Ile44 itself is not actually contacted by the little finger loop (see Fig. 3c), but other hydrophobic residues commonly included in the Ile44 patch are. However, also those are always in the Ubiquitin:USP interface.

These section was removed in line with the previous point. Additionally, we added the information that Ile44 itself is not contacted to the main text.

- It appears as if the little finger loop takes up the exact position of what is commonly referred to as blocking loop 1 (terminology introduced by Hu et al EMBO J 2005, which the authors could also use when referring to canonical USPs). The connectivity in the domain is not equivalent to BL1 though (this could be shown with topology diagrams of a canonical USP and of TSM), but the relative position is. Importantly, BL1 in USPs typically engage the Ile36 patch with a large aromatic residue (e.g., a Trp in USP30 or a Tyr in USP7), however, in TSSM Ubiquitin is rotated such that the Ile44 patch is recognised at the exact spot where normally Ile36 is engaged. Can the authors comment on this interpretation? This reviewer would also find a comparison to the canonical USP recognition mode in the main figures beneficial (however one that is less crowded than in the SI) to make this very important finding come across well. Even if the loop is more like a BL1 (which is highly diverse in the USP family, see reference above), it can still be called "little finger", but the analogies should be carefully phrased throughout the text (e.g., page 9: "functionally but not structurally analogous" would not be correct if it does not replace the fingers which contact Phe4 as BL1 is not part of the fingers).

The reviewer is correct. The positioning of BL1 from canonical USPs and the TssM Littlefinger loop is superficially similar. Since the Littlefinger is derived from a different part of the sequence (Fig 1a) and contacts a different patch of ubiquitin, it is still distinct from the BL1. We have incorporated this information into the main text and generated a new superposition to show the structural similarities (Supplementary Fig 3c). Additionally, we highlighted BL1 and Littlefinger in Fig1a to show the different position within the amino acid sequence. Since BL1 and Littlefinger are distinct from each other we still classify the Littlefinger as functionally (but not structurally) analogous to the fingers domains. Both serve as the main contacts to ubiquitin and (partially) interact with the Ile44 patch, while BL1 contacts the Ile36 patch.

- The assessment of the molecular weight of apo TSSM in solution by SEC is flawed in its current form (page 7, Fig. 4b). There are only two, seemingly randomly chosen, comparison proteins, and the GST-Ub is unsuitable for 36 kDa because it is an obligate dimer (due to the GST) and as a fusion protein has a much higher hydrodynamic radius than a globular protein. There are commercial reference proteins that can be used which have defined oligomerisation states and are (reasonably globular) so that the retention times can be quantitatively analysed. Such a reference, which also includes proteins of smaller sizes, must be used. Alternatively, the authors should use light-scattering (SEC-MALS or SEC-RALS) to unequivocally demonstrate the molecular weight / oligomeric state of TSSM.

The experiment has been re-done using commercially proteins routinely used for the calibration of SEC columns. Additionally, a 75pg column was used to achieve a better resolution of the proteins. The new data confirm the original conclusion that TssM behaves like a monomer in solution.

- The abstract ends with speculation on the presence of the strand-swapping in live cells and of the role of the Ig-fold domain, but not a proper conclusion. The same applies to the ending of the results section (page 8), where it is not clear where the structural analysis of the Ig-like domains is leading. The authors then speculate about sugar binding (page 10) in the discussion. The latter could be substantiated by an analysis of the residues in the possible sugar binding site (if present in their TSSM), and the text be more rounded off at these regions.

We admit that our manuscript does not reveal the function of the Ig-like domain and the strand swap observed in the apo structure. To really address these questions, an infection model would be required, which is outside the scope of our manuscript focussing on the TssM mechanism. As have showed that deletion of the Ig-domain does not alter the DUB properties, the remaining questions cannot be addressed *in vitro*. To address the concerns of the reviewer, we have changed the ending of the result section (page 8) to summarize our *in vitro* data on the Ig-like domain. We have also clarified the paragraph discussing the possibility of sugar binding on pages 10/11. In brief, this speculation was not based on the identification of a possible sugar binding site, but rather on the presumed positioning of the Ig-like domain relative to the bacterial surface and its evolutionary relationship to other sugar-binding domains.

- The authors show in Fig. 1b TSSM reactivity with both Ubiquitin and Nedd8 probes but then qualify the Nedd8-reactivity with the assays shown in Figs. 1cd, which currently looks like it is an issue with the comparison of probe to substrate. However, their probe assay is very long with 18 h. It should be repeated at shorted times (for Ub and Nedd8-PA only) to test if the strong Ub preference seen in the kinetic assay can also be visualized with the probes.

We measured the requested time curve and added the data to the newly generated Supplementary Figure 1. Using very short time points of 3 – 20min, a preferential reactivity with the ubiquitin probe became visible, which supports the AMC results.

- Moreover, the authors prepare Ubl-PA probes by aminolysis of C-terminal thioesters but at very different stoichiometries as reported for Ub-PA (e.g., in Gersch et al. NSMB 2017). The precise amounts should be cross-checked. This reviewer would not be surprised if the conditions of only 2-fold excess of propargylamine over Ubl thioester and the high amount of NaOH led also to hydrolysis of the C-terminus, which by size exclusion chromatography would not be separated. For all probes, intact mass spectra of the used aliquots must be shown to demonstrate the identity of the used reagents. The presence of other species of course does not per se disqualify the assay in Fig. 1b.

Our protocol for generating the Ub-PA probe follows exactly the one described in Gersch et al 2017. While this publication describes the amounts and concentrations of the individual reagents, our description of the protocol documents their final concentration in the reaction mixture. We double-checked the data and found them to be identical. Thus, there is no reason to assume elevated amounts of hydrolysis products. Moreover, the purity of our probes is regularly controlled by intact mass spectrometry (shown below for Ub-PA). The observed ~5% of hydrolysis product should not pose a problem for the assays, which routinely involve a 10x excess of probe over DUB. As documented in the Materials section, the Nedd-PA probe was obtained from Monique Mulder in Leiden.

In addition, the following minor points should be addressed:

- On page 2, the authors state that the 1-2 DUBs typically present in bacteria either target K63-chains or lack specificity (without reference), but on page 10 they state that bacteria typically only have 1 DUB with little specificity. This is contradicting and should be fixed.

We have rephrased the introductory sentence on page 2 and added a reference. The issue on page 10 has been corrected

- On page 3, the authors make the case for TSSM from *B. pseudomallei* by calling it representative. This reviewer would appreciate if the authors could expand on this towards the end of the manuscript and comment on whether they expect the identified Ubiquitin recognition mode to be present also in all (!) other DUB-TSSM proteins, at least all analysed bioinformatically. Importantly, they should include the sequence of TSSM of *B. mallei* (the only in vitro studied TSSM so far) into their analyses in the SI.

We have expanded the paragraph (page 9) where we discuss the conservation of the Littlefinger-based ubiquitin recognition mode. Among the TssM-like DUBs that we identified, only those from other Burkholderia species contain the Littlefinger conservation and are predicted (by alphafold) to use this region for S1-ubiquitin recognition. The two non-Burkholderia TssM-like DUBs (one from *Chromobacterium sinusclupearum*, the other from an unidentified bacterial metagenome) lack the Littlefinger region and the associated ubiquitin recognition mode. This is documented in Supplementary Fig 4. We also added *B. mallei* to the alignment in Supplementary Fig 4a, although it is almost identical to the *B. pseudomallei* sequence.

- On page 4, the authors start with TSSM (193-490) but do not comment on the role of the first 192 residues. What was the rationale for these boundaries?

The N-terminal 192 amino acids are predicted to be unstructured (by alphafold, IUPRED, etc). We therefore decided to use the entire folded part of TssM for the first experiments. The information has been added to the manuscript and a domain scheme was added to the supplementary data (Supplementary Figure 1a).

- On page 4, the authors introduce the use of the fluorogenic AMC substrates with "more quantitative analysis", however, the experiments are rather minimalistic. Only one substrate concentration, and two enzyme concentrations (at least for one substrate) are used, and the data are not really analysed in a quantitative manner. Through curve fitting of the existing data (with fixed restraints) the authors may be able to determine an estimate of the Ub/Nedd8-specificity factor. Moreover, negative controls should be shown in Figure 1c to demonstrate that the Nedd8-curve is above a possible baseline drift.

Following this reviewer's suggestion, we estimated a 116x better cleavage of ubiquitin-AMC by determination of the initial velocities using the linear range of the data presented in Figure 1c.

For ensuring better clarity, we have omitted the negative control data from Fig 1 panel c (comparing Ub-AMC and Nedd8-AMX) but are showing them in panel 1d (comparing Nedd8-AMC to negative control). This figure clearly shows that Nedd8-AMC is really cleaved, whereas no baseline drift is visible.

- On page 4, the authors mention a possible relationship with the Josephin family (which they later disprove through structural comparisons, page 6). It may be helpful to briefly explain the rationale (i.e., why one would even consider a relationship with the Josephin family DUB given the higher homology to the USP fold).

The rationale for considering a relationship to the Josephin family is explained at the end of the introduction section (page 3). We never considered TssM to be closer related to Josephins than to USPs. We rather speculated that the entire USP and Josephin families are distantly related to each other, and that TssM-like USPs might form a kind of missing link. This latter aspect has been disproved (page 6), since the TssM structure is no more Josephin-like than that of any other USP.

- On page 5, the "shortened C-terminus" compared to USP7 is mentioned. This is misleading, as USP7 has an elongated C-terminus compared to most other USPs, and so the TSSM C-terminus is the canonical ending of the USP fold.

The section has been rephrased and now points out that TssM shares the canonical USP C-terminus.

- On page 6, "ubiquitin has multiple specific contacts" - why are the contacts named "specific"?

We have removed the word 'specific'

- On page 10, the TSSM from *C. sinusclupearum* appears without any context. *Chromobacterium* should be spelled out, and it should be discussed if this is the odd one out. It would also be appreciated if the authors could state whether their bioinformatic analysis is comprehensive (i.e., do only the known or all *Burkholderia* strains have TSSM with a DUB profile). And why is there a A0A1J5... sequence included in Supp Fig. 3a without any context?

We agree that this part of the text was confusing. We have now bundled all information on other TssM-like sequences on page 9. There, we explain that TssM-like DUBs are only found in selected *Burkholderia* lineages, and spell out some examples. We also introduce *Chromobacterium sinusclupearum* with its full name and explain that the remaining non-*Burkholderia* TssM homolog is from an unidentified metagenomic sequence. This is also the reason why we refer to this sequence by its Uniprot accession number.

- Some minor polishing in the methods would increase consistency (cloning is only mentioned for pOPIN-K, but TSSM is also expressed from pOPIN-S; mutants are only mentioned for the 292-490 construct, but Fig. 2d shows one in the 193-490 construct; Hampton Additive Screens I-III are likely an internal name and not used by Hampton itself; "different TSSM concentrations (as indicated in the figure legends" are mentioned in the methods, but e.g. in the caption to Fig. 1 no concentration is given).

We have added the pOPIN-S cloning method and corrected the name of the additive screen. We now provide all mutant data in the 292-490 background (Figure 2d was updated accordingly). We made sure that all figure captions mention the enzyme concentrations.

- Likewise, Table 1 needs polishing as commas and points are used interchangeably.

Table 1 has been corrected

- In Fig. 2c, it looks like the catalytic His is built such that there is no hydrogen bonding between Cys and Asp - is there a particular reason for this? If not, the side chain should be flipped so that the nitrogens are positioned for ideal hydrogen bonding.

Fig 2c had been accidentally exported from an early version of the structure. It has now been replaced by a panel that uses the final and deposited version of the structure. Here, the His position is correct.

- In Fig. 3, dotted lines are introduced as "hydrophobic interactions" as per the captions, but some are clearly hydrogen bonds (e.g., from the amide backbone), and for some others one does not see as they emerge from a cartoon.

In Fig 3, dotted lines are not generally introduced as "hydrophobic interactions". Instead, the individual panels have their own definition for the dotted lines. c) hydrophobic d) hydrogen bonds e) hydrophobic.

- In Fig. 4, context should be given to "3VYP" and why it is used here.

We have added an explanation of 3VYP to the figure legend (The reason for showing it is also explained in the main text)

****Referees cross-commenting****

Reviewer comments appear to be fair, balanced and complementary.

Re Reviewer 2's comment on the pre-print: It includes a structure of TSSM bound to ubiquitin (but not of the apo protein). I am not sure if it is appropriate to follow up on the esterase activity. Firstly, there are no tools for it commercially available or readily made in vitro, and secondly it would appear a bit "copycat"-like. Especially since the molecular determinants of what makes a DUB a good esterase are still elusive. A narrower focus of this manuscript, but done very well (also according to what reviewer 3 suggested), might be a more fitting option.

Reviewer #1 (Significance (Required)):

The findings are novel and of high relevance to the broad ubiquitin and bacterial pathogen communities as the study addresses an enigmatic USP-type deubiquitinase which, as the authors reveal, recognizes Ubiquitin in an entirely different way than its eukaryotic counterparts. This is basic research of pronounced conceptual and mechanistic advance, as it demonstrates that the USP domain can be much more diversified than previously assumed. The structural analysis is very thorough, the data are scholarly presented, and interactions/mechanisms carefully validated by mutations.

The strongest point is clearly the structural analysis of Ubiquitin recognition by this extremely truncated USP fold, and the introduction of the little finger motif. The manuscript does not provide cellular validation of the findings, which is fine to this reviewer for the DUB catalysis part. For the part of the N-terminal domain, the manuscript would benefit from a cellular validation or some localisation studies, however, this is beyond what is established in the authors' lab, and therefore has not been asked. This in turn limits the study to a very thorough in-vitro analysis of this DUB for TSSMs with DUB activity, using conventional substrates like polyUb chains and fluorogenic substrates, and providing convincing conclusions.

My expertise lies in DUBs, biochemistry, and structural biology, and I this believe to have sufficient expertise to evaluate the majority of the findings except the bioinformatic algorithms used which are however not an emphasis in this manuscript.

Reviewer #2 (Evidence, reproducibility and clarity (Required)):

In this manuscript, the authors solved crystal structures of apo-TssM and its complex with Ubiquitin. Together with the biochemical assays, the authors highlighted the differences of TssM from other USP family. TssM do not contain the classical finger domain while it has little finger domain that authors defined. The Ile44 patch on the ubiquitin is mainly used for interacting with TssM.

For the clarity of there findings several points should be edited.

1) In Fig 2d, the authors used different constructs for testing catalytic residues. It will be better to be consistent. Though the authors showed that the deletion of Ig-like domain does not affect the catalytic activity of TssM by showing the AMC assay, it does not guarantee that the effect of mutation on catalytic sites is same for both construct.

The experiment has been repeated with TssM²⁹²⁻⁴⁹⁰ C308A and the panel in Fig 2d has been replaced..

2) In fig 2d, authors need to put the label to indicate which linkage of di-Ub is used. Authors did it for figure 2f. Chain type was already stated in the legend, but was now also added to the respective panel.

3) By showing AMC assay (fig 2e) and K63 Ub2 cleavage assay (fig 2f), authors concluded that the deletion of Ig-like domain does not affect the activity of TssM. However, as authors found that the Ig-like domain forms dimer at least in their crystallization condition, one cannot exclude the possibility of the role of Ig-like domain in recognising different ubiquitin chains. I would clarify the words by saying "The Ig-like domain does not affect the cleavage K63-Ub2), or authors can expand the cleavage assay with all the linkages.

We agree with reviewer #2 and rephrased the corresponding sentence.

4) In the apo structure, authors found a domain swapped dimer. One can expect that this dimer is crystallographic artifact and not found in the nature. Indeed, authors could not observe this dimer in the solution when they performed SEC analysis and there was no effect on the catalytic activity when the Ig-like domain is deleted. Because there is no clear evidence of functional importance of this dimer in the manuscript, authors need to clarify about this dimer.

We agree with reviewer #2 that there is no evidence for functional importance of the dimer. We had already addressed this issue in our discussion section, where we hypothesized a potential *in vivo* function. We have now rephrased this section of the discussion in order to make it more precise.

Referees cross-commenting

Agree with Reviewer #1's comment on my points.

Reviewer #2 (Significance (Required)):

The structure of TssM is recently reported in a preprint (<https://doi.org/10.21203/rs.3.rs-2986327/v1>) from

Pruneda (OHSU). In this preprint, they suggested the role of TssM as Ubiquitin esterase which is not explored in the this manuscript. As it is already published and freely available, authors can explore the role of TssM in that direction as well.

Because the preprint do not contain the complex structure of TssM with Ubiquitin, authors can also examine the roles of Ile44 patch-interacting residues on the catalytic activities.

Said preprint contains the TssM~Ub complex structure, but not the structure of the apo form. The role of the Ile44-patch contacting residues is also already analysed in regards of esterase activity. As explained in the 'general points' at the top of the rebuttal letter, the two manuscripts were meant to be submitted simultaneously. Due to a delay in our PDB deposition/validation, we submitted our version two weeks later. Thus, we share the opinion of reviewer #1 that it would be inappropriate to use our competitors' data and analyse esterase activity in our manuscript. Instead, we added a reference to the competing preprint to our discussion section and briefly compare the key findings. The authors of the preprint have agreed to reciprocate and reference our preprint in their revised manuscript.

Also, authors can compare their little finger structure with the preprint.

See point above.

In general, this manuscript is providing several interesting points to the readers working on the ubiquitin, structures of proteins, host-pathogen interactions and especially those who studying deubiquitinases.

Reviewer #3 (Evidence, reproducibility and clarity (Required)):

In this study, Hermanns et al. have examined the Burkholderia TssM deubiquitinase (DUB) for its ubiquitin chain cleavage specificity using in vitro analyses and for a structural rationalization of its specificity by solving crystal structures with and without covalently bound ubiquitin. TssM was previously shown to be a DUB of the USP family capable of cleaving several ubiquitin chain types. From bioinformatic comparisons, the authors inferred that TssM lacks the 'fingers' domain of classic USP enzymes, and this was shown in the co-crystal structure to be replaced by a 'Littlefinger' loop. The work here is well described and appears to be overall technically solid.

My enthusiasm is reduced for several reasons. First, there is no analysis in an (infected) cell model to evaluate the significance of the TssM mutations, for example, the mutations in ubiquitin-interacting surfaces that are not seen in classic USPs. Second, there is very little quantitation of cleavage rates; while I would not demand derivation of kinetic values for all mutants, the qualitative treatment was not always convincing. For example, Y443A was said to cause "strongly reduced cleavage" (Fig. 4h) whereas E378A was said to "only mildly affect cleavage" (Fig. 4i): to my eye, these cleavage rates are only slightly different (2-3 fold?).

In order to support the findings of our gel based assay and to get more quantitative data, we tested all mutants against Ub-AMC. The results are depicted in supplementary figures 3d/f/g and correspond to results of the chain cleavage assays.

Finally, there is a preprint available on Research Square (doi: 10.21203/rs.3.rs-2986327/v1) that shows a potent esterase activity of TssM against ubiquitinated LPS, which is probably its key role in avoiding surveillance and elimination by the host. This paper, although still a preprint, is far more quantitative, includes similar crystallographic data as in the current paper, and describes cellular assays of function. Even without the RS preprint, the Hermanns et al. study provides a fairly modest advance in our knowledge of TssM function; with the preprint, its novelty is, unfortunately, severely reduced.

As explained in the 'general points' at the top of the rebuttal letter, the two manuscripts were meant to be submitted simultaneously. Due to a delay in our PDB deposition/validation, we submitted our version two weeks later. The competing work is currently under review at a top-tier journal and far from being accepted. We therefore ask to consider the significance of our own manuscript based on the peer-reviewed state-of-the-art.

In our revised manuscript, we have added a brief discussion of the pre-published results. The authors of the preprint have agreed to reciprocate and address our data in their revised manuscript.

Minor comments:

Full genus names should be spelled out when they first appear in the text (such as Burkholderia and Chromobacterium).

Genus names are now spelled out.

Fig. 3a is described out of sequence.

A reference to figure 3a was missing in the beginning of the chapter. The reference was added and the figure is now described in sequence.

In Fig 4a, there still seem to be extensive contacts between monomers but the viewing angle could be misleading.

There are no extensive contacts between the two DUB monomers in the right panel (complex structure). We slightly changed the viewing angle in Fig 4a to make this clearer.

****Referees cross-commenting****

I also agree with Reviewer #1's comments

Reviewer #3 (Significance (Required)):

Even without the Research Square preprint (doi: 10.21203/rs.3.rs-2986327/v1), the Hermanns et al. study provides a fairly modest advance in our knowledge of TssM function; but with the preprint, its novelty is, unfortunately, severely reduced.

November 7, 2023

Re: Life Science Alliance manuscript #LSA-2023-02422

Mr. Thomas Hermanns
Institute for Genetics, University of Cologne

Dear Dr. Hermanns,

Thank you for submitting your revised manuscript entitled "The structural basis for deubiquitination by the fingerless USP-type effector TssM" to Life Science Alliance. The manuscript has been seen by the original reviewers whose comments are appended below. While the reviewers continue to be overall positive about the work in terms of its suitability for Life Science Alliance, some important issues remain.

Our general policy is that papers are considered through only one revision cycle; however, given that the suggested changes are relatively minor, we are open to one additional short round of revision. Please note that I will expect to make a final decision without additional reviewer input upon re-submission.

Please submit the final revision within one month, along with a letter that includes a point by point response to the remaining reviewer comments.

To upload the revised version of your manuscript, please log in to your account: <https://lsa.msubmit.net/cgi-bin/main.plex>
You will be guided to complete the submission of your revised manuscript and to fill in all necessary information.

B. MANUSCRIPT ORGANIZATION AND FORMATTING:

Sincerely,

Reviewer #1 (Comments to the Authors (Required)):

The authors have improved the quality of the manuscript through the revision and have incorporated the bulk of the reviewers' suggestions. Both the additional data as well as the changes to the text are appreciated and make the manuscript almost ready for acceptance.

One critical point has remained which is the preparation of the Ubl-PA probes (page 4 of the author response). The authors state where their protocol comes from, that the numbers were double-checked and that the differences are merely due to the absolute amounts vs. concentrations. However, this is not correct.

The authors state in their methods "To synthesize the propargylated probe, 300 μ M Ub/Ubl-MesNa were reacted with 600 μ M propargylamine hydrochloride (Sigma Aldrich) in buffer A containing 150 mM NaOH for 3 h at RT."

The mentioned 2017 paper states "0.7 g propargylamine hydrochloride [...] was dissolved in 7 mL buffer D [...], supplemented with 490 μ L 4 M sodium hydroxide, and added to 6 mL 600 μ M Ub(1-75)-MesNa in buffer D."

This equates to a total volume of 13,5 mL, to a Ub protein concentration of $600 \cdot 6 / 13,5 = 267 \mu\text{M}$ so approx. the 300 μ M of the authors but to a propargylamine hydrochloride (molecular weight: 91,54 g/mol) concentration of $(0,7/91,54) \text{ mol} / 13,5 \text{ mL} = 566 \text{ mM}$ (millimolar, not micromolar), so about 1000x higher than the 600 μ M which the authors state.

The authors are therefore asked to check their numbers again, to cite the appropriate paper if this protocol was used and to provide the missing characterization spectra for the utilized Nedd8, ISGct, Sumo1 and Sumo2 probes (either through referencing previous papers of their or their collaborator's group where such data for the chemically identical probes was included or through inclusion of the spectra in the author response as they have done for Ub-Pa), as per the original revision report.

This may seem like an irrelevant detail, however, since this paper focuses on the in vitro characterization of TssM and since the characterization of TssM's Ub/Ubl specificity is entirely built on the use of these reagents, this reviewer believes that the utilized tools must be characterized sufficiently so that the derived data becomes reliable. I agree with the authors that a small amount of hydrolysis product per se does not pose a problem for the assays, but it is still essential to demonstrate that the required PA-functionalized species are present in substantial amounts since the propensity of different Ubls for C-terminal functionalization varies considerably.

Reviewer #2 (Comments to the Authors (Required)):

This manuscript is pre-evaluated through the review commons. The authors now addressed all the raised questions from reviewers and I recommend this manuscript for the publication in LSA.

Reviewer #1 (Comments to the Authors (Required)):

The authors have improved the quality of the manuscript through the revision and have incorporated the bulk of the reviewers' suggestions. Both the additional data as well as the changes to the text are appreciated and make the manuscript almost ready for acceptance.

One critical point has remained which is the preparation of the Ubl-PA probes (page 4 of the author response). The authors state where their protocol comes from, that the numbers were double-checked and that the differences are merely due to the absolute amounts vs. concentrations. However, this is not correct.

The authors state in their methods "To synthesize the propargylated probe, 300 μ M Ub/Ubl-MesNa were reacted with 600 μ M propargylamine hydrochloride (Sigma Aldrich) in buffer A containing 150 mM NaOH for 3 h at RT.". The mentioned 2017 paper states "0.7 g propargylamine hydrochloride [...] was dissolved in 7 mL buffer D [...], supplemented with 490 μ L 4 M sodium hydroxide, and added to 6 mL 600 μ M Ub(1-75)-MesNa in buffer D.". This equates to a total volume of 13,5 mL, to a Ub protein concentration of $600 \cdot 6 / 13,5 = 267 \mu\text{M}$ so approx. the 300 μ M of the authors but to a propargylamine hydrochloride (molecular weight: 91,54 g/mol) concentration of $(0,7/91,54) \text{ mol} / 13,5 \text{ mL} = 566 \text{ mM}$ (millimolar, not micromolar), so about 1000x higher than the 600 μ M which the authors state.

The calculations of Reviewer #1 are correct. The huge difference of the PA concentration is caused by a typo in the manuscript. We indeed use 600 mM and not μ M PA during the synthesis. The typo was corrected. Additionally, we provide quality control data for all our probes, to show that all probes are the desired product (see next point).

The authors are therefore asked to check their numbers again, to cite the appropriate paper if this protocol was used and to provide the missing characterization spectra for the utilized Nedd8, ISGct, Sumo1 and Sumo2 probes (either through referencing previous papers of their or their collaborator's group where such data for the chemically identical probes was included or through inclusion of the spectra in the author response as they have done for Ub-Pa), as per the original revision report.

As requested we added intact mass spectrometry data for all probes (shown below) and referenced papers for the Nedd8 probe, made by a collaborating group. The probes correspond to their expected monoisotopic masses and show only a very small amount of hydrolysis byproduct.

Expected monoisotopic masses:

3XFLAG-ISG15⁷⁶⁻¹⁵⁶-PA: 11628.7

3XFLAG-Sumo1-PA: 13946.6

Sumo2-PA: 10450.2

Biotin-PEG-Nedd8-PA: 9035

This may seem like an irrelevant detail, however, since this paper focuses on the in vitro characterization of TssM and since the characterization of TssM's Ub/Ubl specificity is entirely built on the use of these reagents, this reviewer believes that the utilized tools must be characterized sufficiently so that the derived data becomes reliable. I agree with the authors that a small amount of hydrolysis product per se does not pose a problem for the assays, but it is still essential to demonstrate that the required PA-functionalized species are present in substantial amounts since the propensity of different Ubls for C-terminal functionalization varies considerably.

The data is provided. See point above.

Reviewer #2 (Comments to the Authors (Required)):

This manuscript is pre-evaluated through the review commons. The authors now addressed all the raised questions from reviewers and I recommend this manuscript for the publication in LSA.

November 27, 2023

RE: Life Science Alliance Manuscript #LSA-2023-02422R

Mr. Thomas Hermanns
University of Cologne
Institute for Genetics
Zùlpicher Straße 47a
Cologne 50674
Germany

Dear Dr. Hermanns,

Thank you for submitting your revised manuscript entitled "The structural basis for deubiquitination by the fingerless USP-type effector TssM". We would be happy to publish your paper in Life Science Alliance pending final revisions necessary to meet our formatting guidelines.

- please provide a clean version of the manuscript file without tracking changes
- please move your main, supplementary figure, and table legends to the main manuscript text after the references section
- we encourage you to revise the figure legend for Figure 4 so that the figure panels are introduced alphabetically. Please correct the figure and its callouts in the manuscript text
- The table can be included at the bottom of the main manuscript or sent separately
- please consult our manuscript preparation guidelines <https://www.life-science-alliance.org/manuscript-prep> and make sure your manuscript sections are in the correct order

A. FINAL FILES:

B. MANUSCRIPT ORGANIZATION AND FORMATTING:

Sincerely,

November 29, 2023

RE: Life Science Alliance Manuscript #LSA-2023-02422RR

Mr. Thomas Hermanns
University of Cologne
Institute for Genetics
Zùlpicher Straße 47a
Cologne 50674
Germany

Dear Dr. Hermanns,

Thank you for submitting your Research Article entitled "The structural basis for deubiquitination by the fingerless USP-type effector TssM". It is a pleasure to let you know that your manuscript is now accepted for publication in Life Science Alliance. Congratulations on this interesting work.

DISTRIBUTION OF MATERIALS:

Again, congratulations on a very nice paper. I hope you found the review process to be constructive and are pleased with how the manuscript was handled editorially. We look forward to future exciting submissions from your lab.

Sincerely,
